# Radiation correction and uncertainty evaluation of RS41 temperature sensors by using an upper-air simulator

Sang-Wook Lee[1,3], Sunghun Kim[1], Young-Suk Lee[1], Byung Il Choi[1], Woong Kang[1], Youn Kyun Oh[1], Seongchong Park[1], Jae-Keun Yoo[1], Joohyun Lee[2], Sungjun Lee[1], Suyong Kwon[1,3], Yong-Gyoo Kim[1]

[1]Division of Physical Metrology, Korea Research Institute of Standards and Science, Daejeon 34113, Republic of Korea
[2]Interdisciplinary Materials Measurement Institute, Frontier of Extreme Physics, Korea Research Institute of Standards and Science, Daejeon 34113, Republic of Korea
[3]Department of Science of Measurement, University of Science and Technology, Daejeon 34113, Republic of Korea

*Correspondence to*: Y.-G. Kim (dragon@kriss.re.kr)

**Abstract.** An upper-air simulator (UAS) has been developed at the Korea Research Institute of Standards and Science (KRISS) to study the effects of solar irradiation of commercial radiosondes. In this study, the uncertainty of the radiation correction of a Vaisala RS41 temperature sensor is evaluated using the UAS at KRISS. First, the effects of environmental parameters including the temperature ($T$), pressure ($P$), ventilation speed ($v$), and irradiance ($S$) are formulated in the context of the radiation correction. The considered ranges of $T$, $P$, and $v$ are –67 to 20 °C, 5−500 hPa, and 4−7 m·s$^{-1}$, respectively, with a

fixed $S_0 = 980$ W·m$^{-2}$. Second, the uncertainties in the environmental parameters determined using the UAS are evaluated to calculate their contribution to the uncertainty in the radiation correction. In addition, the effects of rotation and tilting of the sensor boom with respect to the irradiation direction are investigated. The uncertainty in the radiation correction is obtained by combining the contributions of all uncertainty factors. The expanded uncertainty associated with the radiation-corrected temperature of the RS41 is 0.17 °C at the coverage factor $k = 2$ (approximately 95 % confidence level). The findings obtained

by reproducing the environment of the upper air by using the ground-based facility can provide a basis to increase the measurement accuracy of radiosondes within the framework of traceability to the International System of Units.

# 1 Introduction

The measurement of temperature and humidity in the free atmosphere is of significance for weather prediction, climate monitoring, and aviation safety assurance. Radiosondes are telemetry devices that include various sensors to perform in situ measurements and transmit the measured data to a ground receiver while the device is carried by a weather balloon to an altitude of approximately 35 km. Since their development in the 1930s, radiosondes have been widely used to measure various essential climate variables (ECVs) such as the temperature, water vapour, pressure, wind speed, and wind direction in the upper-air atmosphere. Owing to their high accuracy of 0.3 to 0.4 K claimed by manufacturers (Vaisala), radiosonde measurements provide reference for other remote sensing techniques such as those based on satellite and lidar. However, evaluation methods for their sensor accuracy are not fully disclosed to users. Operation principle of laboratory setups, algorithms to correct measurement errors, and corresponding uncertainty evaluations are prerequisites for a reference data product. The dependence of accuracy evaluation based only on manufacturer data may lead to inhomogeneities in data records due to the use of different radiosonde models.

To ensure the quality control of measurements in the upper air, the Global Climate Observing System (GCOS) Reference Upper-Air Network (GRUAN) was founded in 2008. The key objective of the GRUAN is to perform high quality measurements of selected ECVs from the surface to the stratosphere to monitor climate change. To this end, the required temperature measurement accuracy in the troposphere and stratosphere has been specified as 0.1 K and 0.2 K, respectively (Gcos, 2007).

The main source of error in the temperature measured by radiosondes is solar radiation during sounding in daytime. The temperature sensors of most commercial radiosondes are exposed to solar radiation, which leads to radiative heating of the temperature sensor. According to the last intercomparison of high quality radiosonde systems (Nash et al., 2011), radiation correction values applied by manufacturers were distributed from 0.6 to 2.3 K at 10 hPa. More recently, according to the radiation correction of Vaisala RS41 (Vaisala), it is increased from 0.53 to 1.16 K at 10 hPa as solar angle is elevated from 0° to 90°. Correcting the radiation effect is challenging because the temperature of sensors is also affected by other thermal exchange processes such as conduction from the sensor boom, convective cooling by air ventilation, and long-wave radiation from sensors. To minimize the effect of radiative heating of radiosonde temperature sensors, the size of sensors has been reduced (De Podesta et al., 2018) and highly reflective coatings are used (Luers and Eskridge, 1995; Schmidlin et al., 1986). Moreover, the sensor boom has been redesigned to reduce the thermal conduction to sensors. Nevertheless, the effect of solar irradiation cannot be eliminated and thus should be corrected properly.

Many researchers have attempted to correct the radiation effect on radiosonde temperature sensors through theoretical and experimental techniques. The early theoretical approaches were based on heat transfer equations governing the thermal exchange between the sensor and surrounding media (Luers, 1990; Mcmillin et al., 1992). However, the application of these approaches requires complete knowledge regarding the material properties of the sensor and sensor boom and air

characteristics in a wide range of temperatures, and the aerodynamic characteristics for a specific sensor geometry must be determined.

A few researchers performed in-flight experiments to derive a formula to correct the radiation effect. Radiation correction was estimated by using radiosondes equipped with four thermistors having coatings with different spectral responses, i.e., emissivities and absorptivities (Schmidlin et al., 1986). A correction formula was derived by establishing the relationship

between the irradiance and increase in the temperature via radiative heating during daytime sounding (Philipona et al., 2013). Two identical thermocouples were used to measure the temperature difference when only one sensor was exposed to solar radiation and the other was shielded. As a result, radiation correction was obtained by a linear function of geopotential height which gives 1 K at 32 km. However, the effect of the shield could not be eliminated.

Other groups adopted a chamber system for radiation correction by simulating the upper-air environments including the solar

radiation. The GRUAN conducted experiments by using a chamber that could imitate the pressure, air ventilation, and solar irradiance by using a vacuum pump, fan, and lamp or sunlight, respectively (Dirksen et al., 2014). Recently, the same group conducted experiments by using a new laboratory setup including a wind tunnel with various functionalities and improved uncertainties in processing the GRUAN data for the Vaisala RS41 sensors (Von Rohden et al., in review, 2021). However, these experiments were conducted at room temperature, and thus, the influence of the ambient temperature on the radiation

error was not investigated. Notably, a previous study based on a chamber system reported that the solar-irradiation-induced temperature rise of sensors increases as the air temperature is decreased (Lee et al., 2018a).

Recently, the Korea Research Institute of Standards and Science (KRISS) developed an upper-air simulator (UAS) that can simultaneously control the temperature, pressure, air ventilation, and irradiation (Lee et al., 2020). This UAS has been also used to calibrate the relative humidity sensors of commercial radiosondes at low temperatures (down to −67 °C) (Lee et al.,

75  2021).

In this study, the uncertainty in the radiation correction of a Vaisala RS41 temperature sensor is evaluated using the UAS developed at KRISS (Lee et al., 2020). It is shown how the uncertainty of each environmental parameter and radiosonde movements in the UAS contributes to the uncertainty of RS41 through a radiation correction formula obtained by a series of radiation experiments. The layout of the UAS is described in Section 2, along with the addition of new functions to consider

the effect of the rotation and tilting of the sensor that are an important progress from the previous version of the UAS. As described in Section 3, a radiation correction formula for the RS41 sensor is derived through a series of experiments involving varying temperature ($T$), pressure ($P$), and ventilation speed ($v$) values in the following ranges: –67 to 20 °C, 5−500 hPa, and 4−7 m·s$^{-1}$, respectively, with a fixed irradiance $S_0 = 980$ W·m$^{-2}$. The effects of sensor rotation and tilting with respect to the incident irradiation are also investigated. Section 4 describes the evaluation of the uncertainties associated with the

environmental parameters and sensor motions/positions controlled in the UAS to calculate the contribution of these factors to the uncertainty in the radiation correction. This study can help enhance the measurement accuracy of radiosondes within the

framework of traceability to the International System of Units (SI) by providing a methodology for radiation correction in an environment similar to that which may be encountered by radiosondes.

## 2 Layout of the UAS

### 2.1 Temperature control of the radiosonde test chamber by using a climate chamber

Figure 1(a) shows the test chamber of the UAS with an installed radiosonde for the radiation correction. The test chamber is inside a climate chamber (Tenney environmental, Model: C64RC) of which working space is 1219 mm × 1219 mm × 1219 mm. The temperature of the test chamber is controlled by the climate chamber. Air is precooled before entering into the climate chamber by passing through a heat exchanger in a separate bath (Kambic metrology, Model: OB-50/2 ULT) of which temperature is lower than that of the climate chamber by about 5 °C. The temperature of the precooled air is then adjusted to that of the climate chamber while passing through the second heat exchanger (9.3 m in length) in the climate chamber before entering into the test chamber. The radiosonde is installed upside-down, as shown in Fig. 1(b), and the air flows into the test chamber from the bottom. The temperature of the test chamber is measured using a calibrated platinum resistance thermometer (PRT).

### 2.2 Pressure and ventilation speed control through sonic nozzles and a vacuum pump

To control the air ventilation speed at low pressures, sonic nozzles, also known as critical flow Venturi, are used. The sonic nozzles are fabricated as toroidal-throat Venturi nozzles to comply with the ISO 9300 standard (Iso, 2005) and calibrated using low-pressure gas flow standard system at KRISS (Choi et al., 2010). Thus, the reference value and SI traceability of the ventilation speed are obtained by using the sonic nozzles in the UAS. Sonic nozzles can be used to achieve a specific maximum constant flow when the ratio of the downstream pressure ($P_e$) to the upstream pressure ($P_o$) is smaller than a certain critical point ($P_e/P_o < P_c/P_o$). The test chamber lies in the downstream region of the sonic nozzles, in which the pressure is lowered using a vacuum pump (WONVAC, Model: WOVP-0600) to attain the critical condition. Six sonic nozzles with different throat diameters are used to generate air ventilation speeds ranging from 4 m·s$^{-1}$ to 7 m·s$^{-1}$ in the pressure range of 5–500 hPa. The generated air flow is measured through laser Doppler velocimetry (LDV) (Dantec, Model: BSA F60) to investigate the spatial gradient in the test chamber. Ar-ion laser (3W) having a wavelength of 514.5 nm is used for the LDV with a focal length of 400.1 mm and nominal beam spacing of 33 mm.

### 2.3 Irradiation control by using a solar simulator

Solar irradiation is imitated by using a solar simulator with a xenon DC arc lamp (Newport, Model: 66926-1000XF-R07). The virtual sunlight is irradiated onto the radiosonde temperature sensor and the sensor boom through quartz windows of the test chamber. A constant irradiance of 980 W·m$^{-2}$ at the position of the radiosonde sensors inside the test chamber is adopted

throughout this study. The two-dimensional distribution of the irradiance is recorded at the radiosonde sensor location by using a calibrated Si photodiode (Thorlabs, Model: SM05PD2A). The spatial uniformity of the irradiance around the sensor position is within ±5 %. In addition, the irradiance is monitored to check its drift during the experiments by using a photodiode-based pyranometer (Apogee, Model: SP-110-SS) installed behind the test chamber. The pyranometer is calibrated at KRISS and the uncertainty is 1 % of the measured value with a coverage factor $k = 1$.

## 2.4 Installation of RS41

The uncertainty associated with the radiation correction for a commercial radiosonde (Vaisala, RS41) is evaluated using the UAS. A complete RS41 unit including the sensor boom, antenna, and main body is installed upside-down in the test chamber, as shown in Figs. 1(a) and (b). The sensor boom is placed parallel to the air flow (blue dashed arrows). The sensor boom is irradiated (red dotted arrows) by the solar simulator in a perpendicular manner through quartz windows (50 mm × 70 mm). The temperature recorded by the RS41 is collected through remote data transmission as in the case of soundings by the Vaisala sounding system MW41. Radiation correction by the manufacturer is applied only during the sounding state. The RS41 unit remains at the pre-sounding state in the manual sounding mode throughout the data acquisition, and thus, raw temperature with no radiation correction is obtained.

## 2.5 Rotation and tilting of the sensor boom

A radiosonde exhibits continuous movements such as pendulum and rotational motions during sounding. The geometry of the temperature sensor of the Vaisala RS41 is a rod shape and thus the rotation and tilt affect the effective irradiance and the direction of air ventilation. Other radiosondes using spherical bead thermistors would be less affected by the rotation and tilt. Thus, the angle of the sensor boom with respect to the radiation direction or air flow may constantly vary. To consider this aspect, the UAS is modified to be able to simulate these situations through rotating and tilting of the sensor boom in the test chamber. Figures 1(c)–(e) illustrate the mechanisms in the test chamber that enable the (d) rotation of the radiosonde around the vertical axis and (e) tilting of the sensor boom from the (c) normal position. The rotation cycle and tilt are controlled using stepper motors. Rotation cycles of 5 s, 10 s, and 15 s are employed. The maximum tilt is 27° with respect to the vertical axis. Effects of the rotation and incident angle of irradiation are studied and incorporated in the uncertainty evaluation of the radiation correction of the sensor.

## 3 Experiment Details

### 3.1 Effect of pressure

Radiation error is the temperature difference between the sensor with irradiation and air ($T_{on} - T_{air}$). However, the air temperature measured in the current chamber system does not represent that in free atmosphere since the air is heated by

irradiation for a short time while passing through the test section. It is difficult to measure true air temperature at a shaded area in the test chamber using an independent thermometer because the test section is also slightly heated by the irradiation. The temperature measured below the window is continuously increased by a few tens of mK while repeating the experiments for 10 min. Thus, the radiation correction value ($\Delta T_{rad}$) is obtained by the difference in the temperatures with irradiation ($T_{on}$) and without irradiation ($T_{off}$) as previously reported (Lee et al., 2020); $\Delta T_{rad} = T_{on} - T_{off}$. The duration of irradiation is 120 s and

the measurement is repeated three times.

It has been reported that $\Delta T_{rad}$ significantly increases as the pressure ($P$) decreases from 100 hPa to 7 hPa in the UAS (Lee et al., 2020). In this study, the pressure range is extended (5−500 hPa) to formulate the corresponding effect at a more practical scale. Figures 2(a) shows $\Delta T_{rad}$ as a function of pressure from 5 hPa to 500 hPa with varying temperature ($T$) from −67 °C to 20 °C. The data represents the mean and the standard deviation of three repeated measurements on a single RS41 unit. The

biggest standard deviation was 0.014 °C. The enhanced increase of $\Delta T_{rad}$ is observed at low pressures for all measured temperatures because the convective cooling process is weakened as the air density decreases at low pressures. The effect of temperature is well distinguished in the low-pressure range (5 to 50 hPa), whereas it is not clearly observable in the high pressure range (100 to 500 hPa). This phenomenon can be attributed that the uncertainty of $\Delta T_{rad}$ becomes relatively larger with respect to $\Delta T_{rad}$ as $\Delta T_{rad}$ is decreased at high pressures in the UAS.

To parameterize a radiation correction formula in terms of $T$ and $P$, $\Delta T_{rad}$ at each temperature is fitted individually by using an empirical polynomial function of $\text{Log}_{10} P$, as indicated by dashed lines in Fig. 2(a). The fitting equations represented in Fig. 2(a) are as follows:

$$\Delta T_{rad} = A_0(T) + B_0(T) \cdot \log(P) + C_0(T) \cdot [\log(P)]^2 \quad \text{for } 5 \text{ hPa} \leq P \leq 500 \text{ hPa}, S_0 = 980 \text{ W·m}^{-2}, \tag{1}$$

where $A_0(T)$, $B_0(T)$, and $C_0(T)$ are fitting coefficients with functions of $T$, having units of °C, °C·[log hPa]$^{-1}$, and °C·[log hPa]$^{-2}$, respectively. The irradiation intensity $S_0$ is set as 980 W·m$^{-2}$ throughout this study.

## 3.2 Effect of temperature

The following $T$ values are used in the test chamber: −67 °C, −55 °C, −40 °C, −20 °C, 0 °C, and 20 °C. As shown in Fig. 2(a), $\Delta T_{rad}$ gradually increases as the temperature reduces, especially in the low pressure range of 5−50 hPa. To incorporate the temperature effect in Eq. (1), the coefficients are fitted with empirical linear functions, as follows:

$$A_0(T) = a_0 \cdot T + a_1, \tag{2}$$
$$B_0(T) = b_0 \cdot T + b_1, \tag{3}$$
$$C_0(T) = c_0 \cdot T + c_1, \tag{4}$$

where $a_0$, $a_1$, $b_0$, $b_1$, $c_0$, and $c_1$ are fitting coefficients. Information regarding these coefficients is summarized in **Table 1**.

**Table 1**. Coefficients in Eqs. (2), (3), and (4).

| Coefficient | Unit | Value |
|:---:|:---:|:---:|
| $a_0$ | | $-3.69 \times 10^{-3}$ |
| $a_1$ | °C | $1.25$ |
| $b_0$ | $[\log hPa]^{-1}$ | $2.84 \times 10^{-3}$ |
| $b_1$ | °C·$[\log hPa]^{-1}$ | $-5.98 \times 10^{-1}$ |
| $c_0$ | $[\log hPa]^{-2}$ | $-5.38 \times 10^{-4}$ |
| $c_1$ | °C·$[\log hPa]^{-2}$ | $8.66 \times 10^{-2}$ |

The residuals obtained using Eq. (1) and the associated fitting coefficients listed in **Table 1** are presented in Fig. 2(b). The fitted values agree with the measurement data within ±0.03 °C.

In order to understand the observed temperature effect theoretically, the temperature sensor is modelled as a sphere made of Platinum (Pt) with a diameter ($D$) of 1 mm. The Pt sphere is placed in the middle of an air flow ($v$) with varied temperature ($T_a$) and pressure ($P_a$) as shown in Fig. 3(a). The sphere is heated by the absorption of the solar irradiance ($S = 1000$ W·m$^{-2}$) and cooled by the forced air convection (5 m·s$^{-1}$) similar to the experiment. The radial and angular temperature distribution of the sphere is neglected and assumed to be uniform. Then, the steady state temperature of the sphere ($T_s$) is simply decided by the energy balance of the heat transfer exchange as follows:

$$\alpha S = h(T_s - T_a) \text{ with } h = \frac{k}{D}[2 + (0.4Re^{\frac{1}{2}} + 0.06Re^{\frac{2}{3}})\left(\frac{\mu C_p}{k}\right)^{2/5}] \quad , \tag{5}$$

where $\alpha$ is the absorptivity of the metal sphere, $S$ is the solar irradiance and $h$ is the heat transfer coefficient (Incropera and Dewitt, 2002; Luers and Eskridge, 1995). The net heat transfer by longwave radiation from the Pt sphere is not considered because it is negligible ($\sim 10^{-6}$ W) compared to that by the convective heat transfer in Eq. (5). The heat transfer coefficient $h$ is determined by several parameters concerning the diameter of the sphere ($D$) and the properties of air including thermal conductivity ($k$), viscosity ($\mu$), heat capacity ($C_p$) and Reynolds number ($Re = \rho v D/\mu$), in which $\rho$ and $v$ is the density of air and wind speed, respectively.

The radiation correction ($T_s - T_a$) at $T_a = 20$ °C and $-70$ °C is calculated by Eq. (5) and displayed together with the experimental values (mean and standard deviation of three repeated experiments) as shown in Fig. 3(b). The properties of air (N$_2$) used for the calculation refer to the NIST Chemistry WebBook (Linstrom and Mallard, 2001). In general, the calculated radiation correction of the Pt sphere is elevated as the pressure decreases as in the case of the experiment. This is because the heat transfer coefficient is reduced by about 35 % as the density of air is decreased with varying pressure from $P_a = 50$ hPa to 5 hPa. Interestingly, the temperature effect on the calculated radiation correction is also observed similar to the experiment. The theoretical value is roughly consistent with the experimental value within the uncertainty of $\Delta T_{rad}$ (0.1 °C) as obtained in Section 4.9. A decrease of thermal conductivity of air by about 26 % at $-70$ °C is mainly responsible for the decrease of the

heat transfer coefficient and thereby the increase of the radiation correction at low temperature (−70 °C). The thermal conductivity of air plays an important role for the heat transfer at the boundary between the air and the Pt sphere. The same phenomenon was also observed for thermistors even though there is no apparent air ventilation (Lee et al., 2018a) which may emphasize the role of thermal conductivity of air. The parameters and their values used for the calculation of radiation correction at $T_a$ = 20 °C and −70 °C with $P_a$ = 5 hPa is summarized in **Table 2**.

**Table 2**. Parameters and their values in Eq. (5) and the calculation of radiation correction at 20 °C and −70 °C.

| Parameter | Symbol (Unit) | Value ($T_a$ = 20 °C) | Value ($T_a$ = −70 °C) |
|---|---|---|---|
| Diameter | $D$ (m) | 0.001 | 0.001 |
| Air pressure | $P_a$ (hPa) | 5 | 5 |
| Wind speed | $v$ (ms$^{-1}$) | 5 | 5 |
| Viscosity | $\mu$ (Pa·s) | 0.00001754 | 0.00001307 |
| Density | $\rho$ (kg·m$^{-3}$) | 0.0057466 | 0.0082925 |
| Thermal conductivity | $k$ (W·m$^{-1}$·K$^{-1}$) | 0.025367 | 0.018869 |
| Heat capacity | $C_p$ (J·kg$^{-1}$·K$^{-1}$) | 1039.6 | 1039.1 |
| Reynolds number | $Re$ | 1.64 | 3.17 |
| **Heat transfer coefficient** | $h$ **(W·m$^{-2}$·K$^{-1}$)** | **63.97** | **51.67** |
| Solar irradiance | $S$ (W·m$^{-2}$) | 1000 | 1000 |
| Absorptivity of metal | $\alpha$ | 0.2 | 0.2 |
| **Radiation correction** | $T_s$ **-** $T_a$ **(K)** | **0.78** | **0.97** |

It was previously observed that the temperature rise of RS92 was initially fast due to the small thermal mass of the sensor and subsequently slow (Dirksen et al., 2014). More recently, the temperature of RS41 oscillated when the radiosonde was rotating under irradiation (Von Rohden et al., in review, 2021). These observations are attributed that the heating of the sensor boom with comparably large area is coupled to the heating of the temperature sensor. Since the conductive heat transfer from the sensor boom is missing in the above theoretical calculation, the comparison in Fig. 3(b) may show the effect of the sensor boom on $\Delta T_{rad}$. Interestingly, the growth of $\Delta T_{rad}$ of the theoretical calculation is less steep than that of the experiment as the pressure is decreased to 5 hPa. This may imply that the heat transfer from the sensor boom becomes significant especially at low pressures.

### 3.3 Estimation of the low temperature effect

The effect of low temperature on $\Delta T_{rad}$ is represented by the ratio (%) of $\Delta T_{rad}$ to the corresponding value at 20 °C ($\Delta T_{rad\_20}$), as shown in Fig. 4(a). The data represents the mean and the standard deviation of three repeated measurements on a single
RS41 unit. The temperature effect ($\Delta T_{rad}/\Delta T_{rad\_20}$) gradually increases as the temperature and pressure decrease. $\Delta T_{rad}/\Delta T_{rad\_20}$ is 119 % at $T = -67$ °C and $P = 5$ hPa. To obtain the information required to estimate the low temperature effect by using only $\Delta T_{rad}$ at 20 °C with varied $P$, ($\Delta T_{rad}/\Delta T_{rad\_20} \times 100$) is fitted with empirical linear functions:

$$\Delta T_{rad}/\Delta T_{rad\_20} \times 100 \ (\%) = D(T) \cdot P + E(T) \tag{6}$$

where $D(T)$, represented in hPa$^{-1}$, and $E(T)$, which is dimensionless, are fitting coefficients with functions of $T$. $D(T)$ and $E(T)$
are fitted by linear functions of $T$, as follows:

$$D(T) = d_0 \cdot T + d_1 , \tag{7}$$

$$E(T) = e_0 \cdot T + e_1 , \tag{8}$$

where $d_0$, $d_1$, $e_0$, and $e_1$ are fitting coefficients. The information regarding these coefficients is summarized in **Table 3**.

**Table 3**. Coefficients in Eq. (7) and (8).

| Coefficient | Unit | Value |
|---|---|---|
| $d_0$ | hPa$^{-1}\cdot$°C$^{-1}$ | $2.74 \times 10^{-3}$ |
| $d_1$ | hPa$^{-1}$ | $-2.69 \times 10^{-2}$ |
| $e_0$ | °C$^{-1}$ | $-0.23 \times 10^{0}$ |
| $e_1$ | | $1.04 \times 10^{2}$ |

The residuals obtained using Eqs. (6), (7), and (8) are represented in Fig. 4(b). The estimated values agree with the measurement data within $\pm 1.5$ % (left y-axis), corresponding to approximately $\pm 0.01$ °C (right y-axis). Using Eq. (6), the radiation correction for low temperatures can be estimated through only the room-temperature measurement. Since the
temperature dependency is weak at higher pressures, there is no need to estimate the low temperature effect at 50−500 hPa and the estimation using Eq. (6) is limited within 5−50 hPa.

### 3.4 Effect of ventilation speed

To investigate the effect of ascending speed of radiosondes, the air ventilation speed ($v$) in the test chamber is systematically varied in the range of 4−7 m·s$^{-1}$. Figure 5(a) shows $\Delta T_{rad}$ as a function of the ventilation speed with the temperature varying
from −67 °C to 20 °C. $\Delta T_{rad}$ decreases as the ventilation speed increases, primarily owing to the enhancement in the convective cooling. Because the pressure is fixed at 50 hPa, the temperature effect is clearly visible in Fig. 5(a). The measurement data at

each temperature are fitted using a linear function (dashed lines) to formulate the effect of the ventilation speed. The slope of the linear functions indicates that an increase of 1 m·s$^{-1}$ in $v$ induces a decrease of 0.02–0.03 °C in $\Delta T_{\text{rad}}$. Figure 5(b) shows $\Delta T_{\text{rad}}$ as a function of the ventilation speed with the pressure varying from 5 hPa to 300 hPa. The measurement data at each pressure are fitted using a linear function (dashed lines). The slopes are distributed from –0.04 °C/(m·s$^{-1}$) to –0.02 °C/(m·s$^{-1}$). Although the effect of the ventilation speed is coupled with the temperature and pressure effects, the coupling represented by the variation of slopes in Figs. 5(a) and (b) is minor in the range of 4−7 m·s$^{-1}$. Therefore, the effect of the ventilation speed can likely be treated as an independent parameter. Thus, the ventilation effect is formulated considering the average slope in Figs. 5(a) and (b), which is –0.027 °C/(m·s$^{-1}$). This result is incorporated into Eq. (1) at $v = 5$ m·s$^{-1}$:

$$\Delta T_{\text{rad}} = A_0(T) + B_0(T) \cdot \log(P) + C_0(T) \cdot [\log(P)]^2 - 0.027 \cdot (v\text{-}5) \text{ for } 5 \text{ hPa} \leq P \leq 500 \text{ hPa}, S_0 = 980 \text{ W·m}^{-2} , \quad (9)$$

The residual obtained by applying Eq. (9) is shown in Fig. 5(c). The fitted values agree with the measurement data within ±0.04 °C. The linear relationship between the ventilation speed and the radiation correction in Eq. (9) is only valid in the range of 4−7 m·s$^{-1}$. When $v$ is higher than 7 m·s$^{-1}$ or lower than 4 m·s$^{-1}$, the formula underestimates the correction value.

### 3.5 Effect of irradiation intensity

The linear relationship between $\Delta T_{\text{rad}}$ and the irradiance ($S$) is confirmed with reference to the existing studies based on theoretical and experimental approaches (Luers, 1990; Mcmillin et al., 1992; Lee et al., 2016). $S$ is independent of $T$, $P$, and $v$. As previously observed, the variation of the other parameters results in a change in only the slope of the linear functions ($h$ in Eq. (5)), and the linearity is not altered (Lee et al., 2018c; Lee et al., 2018b). Because all the experiments performed in this study adopt a fixed $S_0 = 980$ W·m$^{-2}$ and the empirical fitting coefficients are accordingly obtained, the effect of the irradiation intensity can be incorporated into Eq. (9) by using the linear relationship between $\Delta T_{\text{rad}}$ and $S$, as follows:

$$\Delta T_{\text{rad}} = S/S_0 \text{ x } [A_0(T) + B_0(T) \cdot \log(P) + C_0(T) \cdot [\log(P)]^2 - 0.027 \cdot (v\text{-}5)] \text{ for } 5 \text{ hPa} \leq P \leq 500 \text{ hPa}, S_0 = 980 \text{ W·m}^{-2} \quad (10)$$

The radiation correction ($\Delta T_{\text{rad}}$) is then scaled with the actual irradiance ($S$) by the factor of $S/S_0$. Consequently, Eq. (10) considers the radiation correction of the RS41 temperature sensor under simultaneously varying $T$, $P$, $v$, and $S$.

### 3.6 Effect of sensor boom rotation

The spinning motion of radiosondes during sounding is imitated by rotating the radiosonde in the test chamber, as shown in Fig. 1(d). The rotation axis is the temperature sensor itself, not the centre of the boom in this work. Therefore, the temperature sensor only spins on the spot and thus the distance between the sensor and the solar simulator does not change during the rotation. The amplitude of the temperature oscillation is investigated by varying the rotation cycle (5 s, 10 s, and 15 s) under irradiation, as shown in Fig. 6(a). The maximum peak ($T_{\text{on\_max}}$) and minimum peak ($T_{\text{on\_mim}}$) appear alternately during the rotation. The difference between the peaks ($T_{\text{on\_max}} - T_{\text{on\_min}}$) for 5 s duration is (0.01–0.02 °C) which is around the measurement resolution of RS41 (0.01 °C) but is increased with the rotation period. Each peak appears twice in a single cycle,

as clearly observed in the 15 s cycle. The exposed surface of the sensor boom depends on the incidence angle, and passes through a maximum twice during a full rotation. The sensor boom experiences irradiation in the perpendicular and parallel directions at $T_{\text{on\_max}}$ and $T_{\text{on\_min}}$, respectively. This finding suggests that the conductive heat transfer from the boom to the sensor influences $T_{\text{on\_max}}$.

Figure 6(b) shows ($T_{\text{on\_max}} - T_{\text{on\_min}}$) as a function of pressure under different rotation cycles. The pressure effect is clearly visible when the rotation cycle is 15 s. Because the experiment is conducted at $T = 25$ °C and $v = 5$ m·s$^{-1}$, the effect of rotation at the lowest considered temperature (−67 °C) is estimated using Eqs. (6), (7), and (8). At $P = 5$ hPa, the value of ($T_{\text{on\_max}} -$ $T_{\text{on\_min}}$) at −67 °C is 20 % higher than that at 25 °C.

The maximum value of ($T_{\text{on\_max}} - T_{\text{on\_min}}$) in the UAS (0.05 °C) is much smaller than that of von Rohden *et al.* (0.3 °C) (Von Rohden et al., in review, 2021). In the work of von Rohden *et al*, although the distance from the light source to the sensor is constant, that to the sensor boom changes with rotation. This may be the reason why the maximum peak appears once in a full cycle when the sensor boom is close to the light source and the ($T_{\text{on\_max}} - T_{\text{on\_min}}$) is bigger than this work. It should be highlighted that the relatively small ($T_{\text{on\_max}} - T_{\text{on\_min}}$) with respect to $\Delta T_{\text{rad}}$ observed in this work suggests that the contribution of the thermal conduction to $\Delta T_{\text{rad}}$ is small compared to that by the direct irradiation of the sensor.

**3.7 Effect of solar incident angle**

The incident angle of irradiation to sensors primarily depends on the solar elevation angle and, during soundings, may also vary due to pendulum motion of the radiosonde. To investigate the effect of the solar incident angle, the sensor boom is tilted by $\theta$ with respect to the normal direction in the test chamber, as shown in Fig. 1(e). Figure 7(a) shows $\Delta T_{\text{rad}}$ as a function of pressure when the sensor boom is in the normal and tilted ($\theta = 27°$) positions. $\Delta T_{\text{rad}}$ in the tilted position (red circle) is lower than that in the normal position (black square) because the effective irradiance ($S_{\text{eff}}$) is reduced by the tilting ($S_{\text{eff}} = S \times \cos 27°$). Because $\Delta T_{\text{rad}}$ is proportional to $S_{\text{eff}}$, the ratio $\Delta T_{\text{rad\_tiled}}/\Delta T_{\text{rad\_normal}}$ should be cosine 27°. The ratio roughly follows the theoretical value (blue dotted line). However, this value is slightly higher and lower than cosine 27° at pressure values less and more than 50 hPa, respectively. At higher pressures, this deviation can be explained by the effect of ventilation, which intensifies in the case of tilting of the sensor boom. However, the reason for the deviation from the theoretical value at low pressures remains unclear. In this paper, the effect of solar incident angle (or tilt angle, $\theta$) is considered by using $S_{\text{eff}}$ ($S \times \cos \theta$) and thus Eq. (10) is revised into its final form as follows:

$$\Delta T_{\text{rad}} = (S_{\text{eff}}/S_0) \times [A_0(T) + B_0(T) \cdot \log(P) + C_0(T) \cdot [\log(P)]^2 - 0.027 \cdot (v\text{-}5)] \text{ for 5 hPa} \leq P \leq 500 \text{ hPa}, S_0 = 980 \text{ W·m}^{-2}, \quad (11)$$

Figure 7(b) shows the difference between $\Delta T_{\text{rad\_tilted}}$ and $\Delta T_{\text{rad\_normal}} \times \cos 27°$ as a function of the pressure. Because the experiment is conducted at $T = 25$ °C, the effect of the solar incident angle at the lowest considered temperature (−67 °C) is estimated using Eqs. (6), (7), and (8). At $P = 5$ hPa, ($\Delta T_{\text{rad\_tilted}} - \Delta T_{\text{rad\_normal}} \times \cos 27°$) at −67 °C is 20 % higher than that at 25 °C. This value is used for the uncertainty due to the tilting of sensor boom in Section 4.7.

## 4 Uncertainty

### 4.1 Uncertainty factors

The uncertainty factors that contribute to the uncertainty budget of the radiation correction are summarized in **Table 4**, in addition with the experimental ranges considered in this work.

**Table 4**. Uncertainty factors and experimental ranges considered in this work.

| Figure | $T$ (°C) | $P$ (hPa) | $v$ (m·s$^{-1}$) | $S$ (W·m$^{-2}$) | Position/Motion |
|--------|----------|-----------|------------------|------------------|-----------------|
| 2 | −67 to 20 | 5−500 | 5 | 980 | Normal |
| 4(a) | −67 to 20 | 50 | 4−7 | 980 | Normal |
| 4(b) | −40 | 5−300 | 4−7 | 980 | Normal |
| 5 | 25 | 5−500 | 5 | 980 | 360° Rotation |
| 6 | 25 | 5−500 | 5 | 980 | 27° Tilted |

### 4.2 Uncertainty in the temperature, $u(T)$

The temperature of the test chamber is measured using a PRT installed in a shaded area. The PRT is calibrated at KRISS, and the calibration uncertainty is 0.025 °C with the coverage factor $k = 1$. The resistance of the PRT is measured using a digital multimeter calibrated at KRISS. Moreover, the stability of the temperature measured using the PRT is considered in determining $u(T)$. The uncertainty components and their contributions to $u(T)$ are listed in **Table 5**.

**Table 5.** Uncertainty budget for the test chamber temperature.

| Uncertainty component | Contribution (°C) |
|-----------------------|-------------------|
| Calibration of the PRT | 0.025 |
| Calibration of the multimeter | 0.010 |
| Stability of temperature measurement | 0.007 |
| $u(T)$, $k = 1$ | 0.028 |

### 4.3 Uncertainty in the pressure, $u(P)$

The pressure of the test chamber is measured using three pressure gauges for different pressure ranges. The gauges are calibrated at KRISS, and the calibration uncertainty is considered in determining $u(P)$. Moreover, the stability of the pressure measured using each pressure gauge is considered to determine $u(P)$. The uncertainty components and their contributions to $u(P)$ are listed in **Table 6**.

**Table 6.** Uncertainty budget for the test chamber pressure.

| Uncertainty component | Pressure range (hPa) | Contribution (hPa) |
|---|---|---|
| | 0−10 | 0.007 |
| Calibration of the pressure gauge | 10−100 | 0.08 |
| | 100−1000 | 0.1 |
| | 0−10 | 0.005 |
| Stability of pressure measurement | 10−100 | 0.11 |
| | 100−1000 | 0.14 |
| | 0−10 | 0.01 |
| $u(P)$, $k = 1$ | 10−100 | 0.14 |
| | 100−1000 | 0.18 |

### 4.4 Uncertainty in the ventilation speed, $u(v)$

The SI traceability of the ventilation speed in the test chamber of the UAS is ensured by calibrating the sonic nozzles at KRISS. The calibration uncertainty of the sonic nozzles is 0.09 % ($k = 1$). The stability of the ventilation speed in the test chamber is

330 considered to determine $u(v)$. The spatial gradient of the ventilation speed in the test chamber is measured through the LDV at KRISS. The measurement dimension using the LDV was 30 mm x 30 mm around the sensor (central) location with 5 mm interval (49 points). Thus, the outermost measurement points were spaced 10 mm apart from the walls of the test chamber (50 mm x 50 mm). The measurement was performed at the condition of $v$ = 4.67 m·s⁻¹ (reference value), $P$ = 550 hPa, and room temperature. The flow regime is turbulent because Reynolds number is high (~$10^5$) at this experimental condition. The average

and the standard deviation by the LDV over the entire measurement area were 4.63 m·s⁻¹ and 0.47 m·s⁻¹, respectively. Although the flow rate of the outermost points tends to be smaller than others, no significant spatial gradient is observed. This may be because the spacing (10 mm) between the outermost measurement points and the walls of the test chamber. The difference between the reference and the measurement average is assumed to have a rectangular probability distribution for the calculation

of the uncertainty of spatial gradient. Then, the standard uncertainty of this estimate is the half-width of the distribution divided by $\sqrt{3}$ (Iso, 2008). The uncertainty components and their contributions to $u(v)$ are summarized in **Table 7**.

**Table 7.** Uncertainty budget for the ventilation speed in the test chamber.

| Uncertainty component | Contribution ($m \cdot s^{-1}$) |
|---|---|
| Calibration of sonic nozzles | 0.005 |
| Stability | 0.052 |
| Spatial gradient | 0.026 |
| $u(v)$, $k = 1$ | 0.058 |

## 4.5 Uncertainty in the irradiance, $u(S)$

The irradiance in the test chamber is measured using a pyranometer. The pyranometer is calibrated at KRISS, and the calibration uncertainty is 9.8 $W \cdot m^{-2}$ at $k = 1$. The stability of the irradiance measured using the pyranometer is considered to determine $u(S)$. The uncertainty of the solar simulator will be negligible compared to that of the actual radiation field in atmospheric soundings due to the lack of knowledge. In addition, the two-dimensional spatial uniformity of the irradiance in the test chamber is measured by moving the pyranometer. The spatial gradient is within ±5 %, and a rectangular probability distribution is assumed for the uncertainty calculation. The uncertainty components and their contributions to $u(S)$ are summarized in **Table 8**.

**Table 8.** Uncertainty budget for the irradiance in the test chamber.

| Uncertainty component | Contribution ($W \cdot m^{-2}$) |
|---|---|
| Calibration of pyranometer | 9.8 |
| Stability | 6.0 |
| Spatial gradient | 28.3 |
| $u(S)$, $k = 1$ | 30.5 |

## 4.6 Uncertainty due to sensor rotation

Since the sensor boom position for $T_{\text{on\_max}}$ during the rotation corresponds to the normal position, the uncertainty due to sensor rotation is obtained based on ($T_{\text{on\_max}} - T_{\text{on\_min}}$), as shown in Fig. 6(b). The value estimated for $T = -67$ °C and $P = 5$ hPa is used to include sufficient uncertainty. The values are assumed to have a rectangular distribution, and thus, the corresponding

standard uncertainty ($k = 1$) is obtained considering the half-maximum value (0.03 °C) divided by $\sqrt{3}$. The reason of using the

half-maximum is that ($T_{\text{on\_max}} - T_{\text{on\_min}}$) is about double of ($T_{\text{on\_max}} - T_{\text{on}}$) or ($T_{\text{on}} - T_{\text{on\_min}}$). Consequently, the uncertainty due to sensor rotation is 0.017 °C ($k = 1$).

### 4.7 Uncertainty due to tilting of the sensor

The uncertainty due to tilting of sensor boom is obtained using ($T_{\text{on\_tilted}} - T_{\text{on\_normal}} \cdot \cos 27°$) shown in Fig. 7(b). The value estimated for $T = -67$ °C and $P = 5$ hPa is used to include sufficient uncertainty. The values are assumed to have a rectangular

distribution, and thus, the corresponding standard uncertainty ($k = 1$) is obtained considering the maximum value (0.045 °C) divided by $\sqrt{3}$. Consequently, the uncertainty due to sensor rotation is 0.026 °C ($k = 1$).

### 4.8 Uncertainty due to fitting error

Because Eq. (11) is used for the final radiation correction, the residuals shown in Figs. 2(b) and 5(c) must be considered in determining the uncertainty. The residuals are assumed to have a rectangular distribution, and thus, the corresponding standard

uncertainty ($k = 1$) is obtained considering the maximum absolute value divided by $\sqrt{3}$. Consequently, the uncertainty due to the fitting error is 0.023 °C ($k = 1$).

### 4.9 Uncertainty budget for radiation correction

The uncertainties in $T$, $P$, $v$, and $S$ contribute to the uncertainty in the radiation correction by the uncertainty propagation law based on Eq. (11):

$$\frac{\partial \Delta T_{\text{rad}}}{\partial T} \cdot u(T) , \tag{12}$$

$$\frac{\partial \Delta T_{\text{rad}}}{\partial P} \cdot u(P) , \tag{13}$$

$$\frac{\partial \Delta T_{\text{rad}}}{\partial v} \cdot u(v) , \tag{14}$$

$$\frac{\partial \Delta T_{\text{rad}}}{\partial S} \cdot u(S) , \tag{15}$$

where $u(\text{parameter})$ represents the standard uncertainty in each parameter at $k = 1$, and the partial differential terms represent

the sensitivity coefficients. The sensitivity coefficients of the uncertainties due to sensor rotation, tilting of the sensor, and fitting error are 1 because they directly contribute to the uncertainty in the radiation correction. The uncertainty budget for the radiation correction ($\Delta T_{\text{rad}}$) based on the conducted experiments is presented in **Table 9**.

**Table 9.** Uncertainty budget on the radiation correction ($\Delta T_{\text{rad}}$).

| Uncertainty factor | Condition | Unit | Standard uncertainty ($k = 1$) | Contribution to uncertainty of radiation correction ($k = 2$) |
|---|---|---|---|---|

| | | | | |
|---|---|---|---|---|
| $T$ | -67 | °C | 0.028 | 0.000 °C |
| $P$ | 5 | hPa | 0.01 | 0.000 °C |
| $v$ | 5 | m·s$^{-1}$ | 0.058 | 0.004 °C |
| $S$ | 980 | W·m$^{-2}$ | 30.5 | 0.062 °C |
| Rotation | 24 | °·s$^{-1}$ | - | 0.035 °C |
| Tilting | 27 | ° | - | 0.052 °C |
| Fitting error | –0.024 – 0.04 | °C | 0.023 | 0.046 °C |
| Expanded uncertainty of radiation correction ($k$ = 2) | | | | 0.100 °C |


## 4.10 Uncertainty budget for the corrected temperature, $T_{cor}$

The corrected temperature ($T_{cor}$) is obtained by subtracting $\Delta T_{rad}$ from the raw temperature ($T_{raw}$), as follows:

$$T_{cor} = T_{raw} - \Delta T_{rad} \ . \tag{16}$$

Thus, the uncertainty in the corrected temperature, $u(T_{cor})$ is calculated as follows:

$$u(T_{cor})^2 = u(T_{raw})^2 + u(\Delta T_{rad})^2 \ , \tag{17}$$

where $u(T_{raw})$ is the standard uncertainty in the raw temperature ($k$ = 1). The uncertainty in $\Delta T_{rad}$, indicated in **Table 9**, must be rescaled in proportion to the actual solar irradiance for Eq. (17). Therefore, the uncertainty in $\Delta T_{rad}$ is scaled up to a level of solar constant (~1360 W·m$^{-2}$) by a factor of (1360/980) based on the linear relationship between $\Delta T_{rad}$ and $S$.

The calibration uncertainty associated with the temperature sensor must be considered to account for the uncertainty in the raw
temperature, $u(T_{raw})$. Consequently, the expanded uncertainty in the corrected temperature of RS41 is 0.138 °C ($k$ = 2), as indicated in **Table 10**. The calibration uncertainty in the RS41 temperature sensor, $U(T_{raw})$ ($k$ = 2) is specified by the manufacturer (Vaisala). Since Vaisala provides additional uncertainty of reproducibility in sounding (0.15 °C when $P$ > 100 hPa, 0.3 °C when $P$ < 100 hPa) (Vaisala), this should be added to the total uncertainty of the corrected temperature when the radiation correction formula in Eq. (11) is applied to soundings.


**Table 10.** Uncertainty budget for the corrected temperature.

| Uncertainty factor | Uncertainty ($k$ = 2) |
|---|---|
| Expanded uncertainty for the radiation correction at 1360 W·m$^{-2}$, $U(\Delta T_{rad})$ | 0.138 °C |
| Calibration of RS41 temperature sensor (Vaisala), $U(T_{raw})$ | 0.100 °C |
| Expanded uncertainty in the corrected temperature, $U(T_{cor})$ | 0.170 °C |

**4.11 Comparison of RS41 radiation correction specified by Vaisala and that obtained through the UAS**

The radiation correction of RS41 by the UAS is based on Eq. (11) for different pressure ranges. In order to apply the correction
formula to actual soundings, the effective irradiance to the sensor should be known. However, radiosondes constantly change
positions with respect to the solar irradiation through rotation and pendulum motion, the calculation of effective irradiance
resorts on the mean of effective irradiance over the motion of radiosondes. Figure 8(a) shows a schematic diagram of a
radiosonde with parameters that affect the effective irradiance $S_{eff}$ on the sensor. Then, the effective irradiance to the sensor
can be calculated as follows:

$$S_{eff} = S_{dir} \cdot \left| \cos\alpha \, \cos\Theta \, \cos\varphi - \sin\Theta \, \sin\alpha) \right| , \tag{18}$$

$S_{dir}$ is solar direct irradiance, $\theta$ is boom tilting angle, $\alpha$ is solar elevation angle and $\varphi$ is azimuthal angle. The effective irradiation
area ($A_{eff}/A_0$) on the sensor boom is averaged over rotation ($\varphi$) with a fixed tilting angle $\theta = 45°$ and plotted as a function of
the solar elevation angle as shown in Fig. 8(b). Using this effective irradiance, the radiation correction by the UAS is obtained
and compared with that of the manufacturer at two different $\alpha$ (45° and 90°) as shown in Fig. 9. For the UAS correction, the
solar direct irradiance is assumed to be 1360 W·m$^{-2}$ at all pressure values. To simulate the albedo effect, the radiation correction
with additional irradiance of 400 W·m$^{-2}$ is also calculated. Consequently, the radiation correction of the UAS is smaller than
the Vaisala by about 0.5−0.7 °C at −70 °C and 5 hPa when only the solar direct irradiance (1360 W·m$^{-2}$) is considered with
the solar elevation angle $\alpha = 45$−90°. When the albedo effect is additionally included (400 W·m$^{-2}$), the gap between the two
corrections is reduced to 0.04−0.4 °C at −70 °C and 5 hPa with $\alpha = 45$−90°. Since solar direct irradiance (1360 W·m$^{-2}$) and
additional diffuse irradiance (400 W·m$^{-2}$) are applied for all pressures, the radiation correction of this work can be exaggerated
at high pressures. The radiation correction of the UAS is smaller than that of the manufacturer at low pressures, which is
consistent with the recent finding using an independent laboratory setup. In the work of von Rohden *et al.*, the radiation
correction was smaller than the manufacturer's by 0.35 K at 35 km (Von Rohden et al., in review, 2021). The radiation
corrections of the manufacturer and the UAS at some representative conditions are summarized in **Table 11**.

**Table 11**. Radiation correction of RS41 by the manufacturer (Vaisala) and the UAS using Eq. (11).

| Pressure (hPa) | Radiation correction by Vaisala ($v = 6$ m·s$^{-1}$) | | | Radiation correction by the UAS ($v = 6$ m·s$^{-1}$, $S_{dir} = 1360$ W·m$^{-2}$, $\theta = 45°$) | | | |
|---|---|---|---|---|---|---|---|
| | $\alpha = 0°$ | $\alpha = 45°$ | $\alpha = 90°$ | $T = -70$ °C, $\alpha = 0°$ | $T = -70$ °C, $\alpha = 45°$ | $T = -70$ °C, $\alpha = 90°$ | $T = -70$ °C, $\alpha = 90°$ +400 W·m$^{-2}$ |
| 1000 | 0.00 °C | 0.10 °C | 0.11 °C | 0.15 °C | 0.17 °C | 0.24 °C | 0.33 °C |
| 500 | 0.03 °C | 0.17 °C | 0.19 °C | 0.17 °C | 0.19 °C | 0.26 °C | 0.37 °C |
| 200 | 0.09 °C | 0.29 °C | 0.32 °C | 0.21 °C | 0.23 °C | 0.33 °C | 0.47 °C |
| 100 | 0.16 °C | 0.42 °C | 0.45 °C | 0.26 °C | 0.29 °C | 0.41 °C | 0.58 °C |

| 50 | 0.24 °C | 0.58 °C | 0.62 °C | 0.32 °C | 0.36 °C | 0.51 °C | 0.72 °C |
| 20 | 0.39 °C | 0.85 °C | 0.90 °C | 0.43 °C | 0.48 °C | 0.67 °C | 0.95 °C |
| 10 | 0.53 °C | 1.10 °C | 1.16 °C | 0.53 °C | 0.58 °C | 0.82 °C | 1.16 °C |
| 5 | 0.68 °C | 1.39 °C | 1.45 °C | 0.64 °C | 0.71 °C | 1.00 °C | 1.41 °C |

## 5 Conclusions

The UAS developed at KRISS provides a unique opportunity to correct the solar radiation effect on commercial radiosondes by reproducing the environments that may be encountered by radiosondes by simultaneously controlling $T$, $P$, $v$, and $S$. The following ranges of $T$, $P$, and $v$ are considered in this study: –67 °C to 20 °C, 5−500 hPa, and 4−7 m·s$^{-1}$, respectively, with a fixed $S_0 = 980$ W·m$^{-2}$. The functionalities of rotating and tilting the sensor boom are added considering the previous report on the UAS (Lee et al., 2020) to investigate the effect of the radiosonde motions with respect to the solar irradiation direction during ascent. The correction formula for the radiation effect on a Vaisala RS41 temperature sensor is derived through a series of experiments with varying environmental parameters and motions/positions of the radiosonde sensor. In addition, an empirical formula is derived to estimate the low temperature effect by using only the inputs of room-temperature measurements. The uncertainty associated with the radiation correction is evaluated by combining the contribution of each uncertainty factor. The uncertain factors considered for the radiation correction are $T$, $P$, $v$, and $S$ as well as the sensor rotation, sensor tilting, and data-fitting-induced errors. The uncertainty budget for the radiation correction of RS41 temperature sensor is 0.1 °C at $k = 2$. When the uncertainty in the absolute temperature measurement (calibration uncertainty) is included, the uncertainty in the corrected temperature is estimated to be 0.17 °C at $k = 2$. The radiation correction values by the UAS are provided when the solar constant (1360 W·m$^{-2}$) is used for $S$ for the comparison with those by the manufacturer. The radiation correction by the UAS depends on effective solar irradiance. Thus, the measurement of solar irradiance in situ and the calculation of effective irradiance are desirable to reflect the conditions such as clouding, solar elevation angle, and radiosonde movement, thereby obtaining more accurate correction values. To measure the solar irradiance in situ, a radiosonde model using dual temperature sensors with different emissivity values has been already tested using the UAS. The temperature difference in the two sensors of the radiosonde is recorded with varying environmental parameters in the UAS to be reversely used to measure solar irradiance in situ during sounding. In this sense, the approach based on dual sensors is different from previous works that estimate the air temperature using several other temperatures measured by sensors with different emissivity (Schmidlin et al., 1986).

As the UAS can support wired and wireless data acquisition, it can be used for any type of commercial radiosonde to derive the radiation correction along with the corresponding uncertainty. Therefore, the UAS can help enhance the measurement accuracy of commercial radiosondes within the framework of the SI traceability.

*Aknowledgement*

This work was supported by the Korea Research Institute of Standards and Science under the grant GP2021-0005-02.

*Author contribution*

SL analysed the experimental data and wrote the manuscript. SK and YL conducted experiments. BC built the humidity control system, WK and YO built the air flow control system, SP and JY established the solar simulator setup. JL conducted theoretical calculation. SL and SK developed the measurement software. YK designed the experiments.

*Competing interests*

The authors declare that they have no conflicts of interest.

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

**Figure captions**

 **Figure 1**: Photographs of the (a) upper air simulator (UAS) and (b) test section with a radiosonde (Vaisala, RS41). Schematics of the radiosonde in the UAS at (a) normal, (b) rotating, and (c) tilted positions.

**Figure 2**. Temperature rise ($\triangle T_{rad}$) in a RS41 temperature sensor due to irradiation as a function of the air pressure in the range of (a) 5–500 hPa and (b) Residuals as a function of air pressure when Eq. (1) is used.

**Figure 3**. (a) Schematic diagram for calculation of radiation correction on a metal sphere and (b) $\triangle T_{rad}$ of the metal sphere
 obtained by the theoretical calculation using Eq. (5) and the experimental value by the UAS as a function of air pressure at two different temperatures.

**Figure 4**: (a) Effect of temperature on $\triangle T_{rad}$ normalized by that at 20 °C ($\triangle T_{rad\_20} = 100$ %) and (b) residuals of linear fittings as a function of the air pressure.

**Figure 5**. Effect of ventilation speed on $\triangle T_{rad}$ at (a) $P = 50$ hPa at different temperatures and (b) $T = –40$ °C at different air
 pressure values. (c) Residuals as a function of the ventilation speed when Eq. (9) is used.

**Figure 6**. (a) Effect of sensor rotation with varied cycles (5 s, 10 s, and 15 s) at $T = 25$ °C and $v = 5$ m·s$^{-1}$ and (b) difference in the maximum and minimum temperature values ($T_{on\_max} – T_{on\_min}$) as a function of the air pressure. $T_{on\_max} – T_{on\_min}$ at 100 hPa and 5 hPa at –67 °C are estimated using Eqs. (6), (7), and (8).

**Figure 7**. (a) Effect of tilting of the sensor boom showing (left y-axis) $\triangle T_{rad}$ with normal ($\triangle T_{rad\_normal}$) and 27° tilted position
 ($\triangle T_{rad\_tilted}$) and the ratio between them (right y-axis). (b) Residual between $\triangle T_{rad\_tilted}$ and $\triangle T_{rad\_normal} \times \cos 27°$ at $T = 25$ °C and the estimate of the residual at $T = -67$ °C by using Eqs. (6), (7), and (8).

**Figure 8**. (a) Schematic diagram for the calculation of effective solar irradiance to the sensor. $\theta$, $\alpha$ and $\varphi$ are the tilting angle of the sensor boom, solar elevation angle and azimuthal angle, respectively. (b) Effective irradiation area of the sensor boom obtained by the mean over rotating radiosonde ($\varphi$) as a function of solar elevation angle ($\alpha$) with the tilting anlge $\theta = 45°$.

 **Figure 9**. Comparison of the radiation correction value between the Vaisala and the UAS as a function of pressure when the solar elevation angle is (a) $\alpha = 45°$ and (b) $\alpha = 90°$ with a boom tilting angle of $\theta = 45°$.

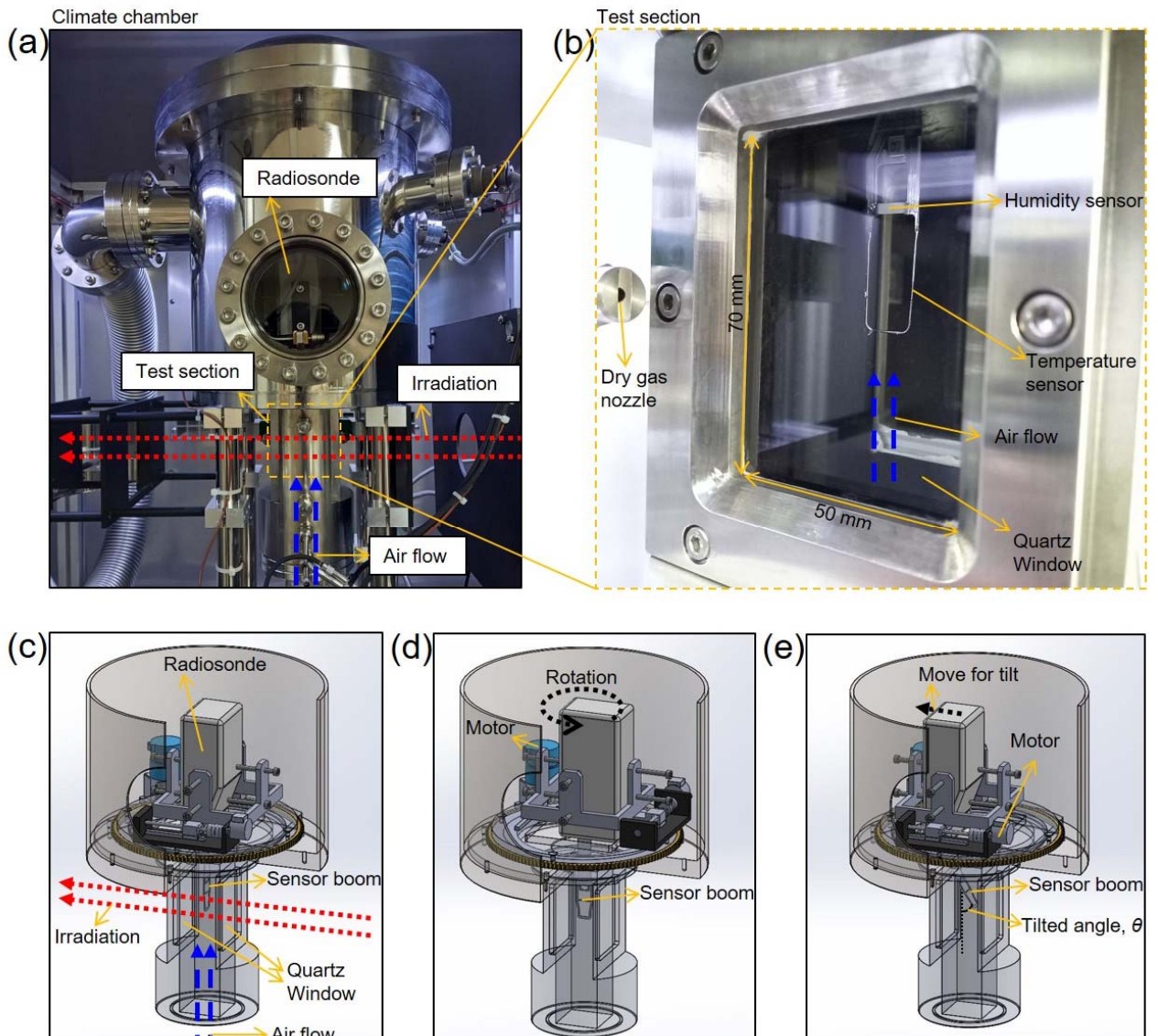

**Figure 1.** Photographs of the (a) upper air simulator (UAS) and (b) test section with a radiosonde (Vaisala, RS41). Schematics

of the radiosonde in the UAS at (a) normal, (b) rotating, and (c) tilted positions.

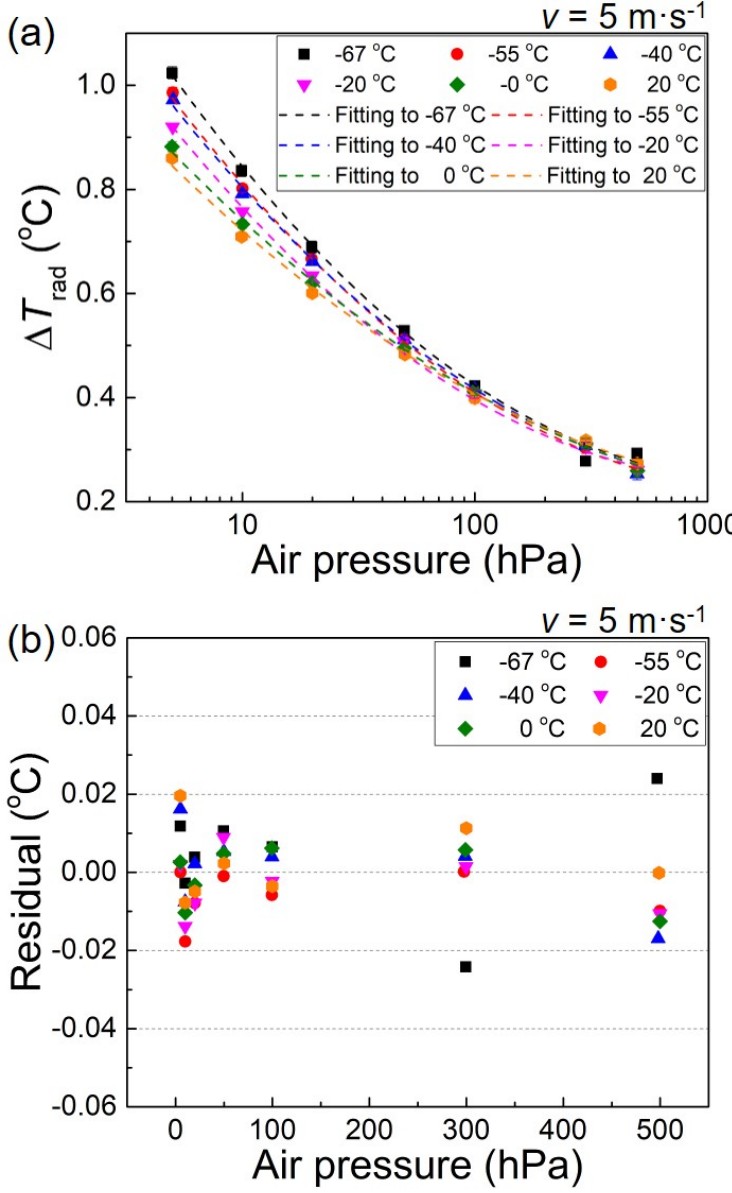

**Figure 2**. Temperature rise ($\triangle T_{rad}$) in a RS41 temperature sensor due to irradiation as a function of the air pressure in the range of (a) 5–500 hPa and (b) Residuals as a function of air pressure when Eq. (1) is used.

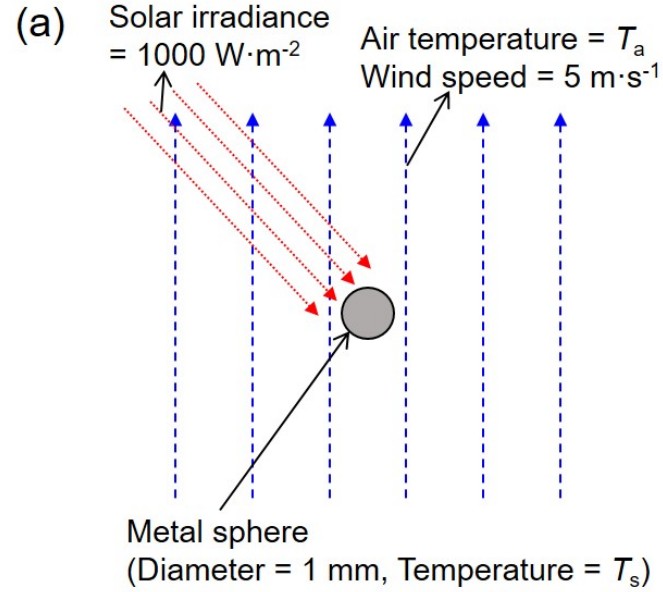

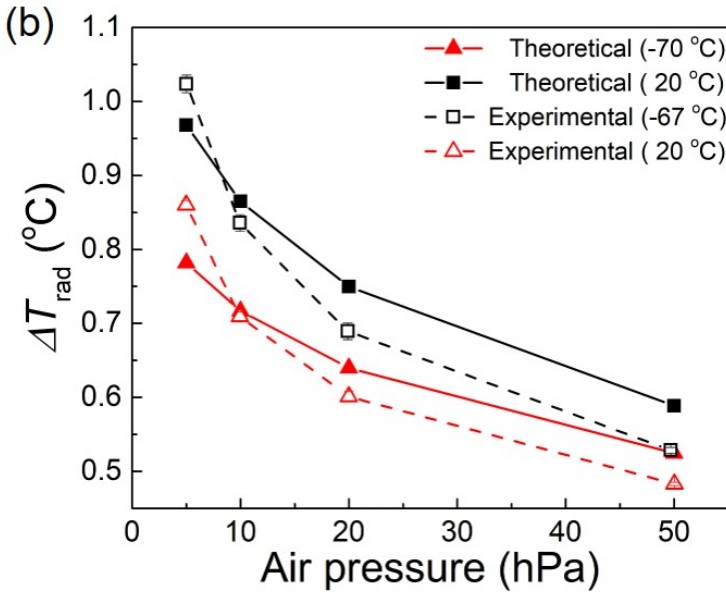

**Figure 3**. (a) Schematic diagram for calculation of radiation correction on a metal sphere and (b) $\triangle T_{rad}$ of the metal sphere obtained by the theoretical calculation using Eq. (5) and the experimental value by the UAS as a function of air pressure at two different temperatures.

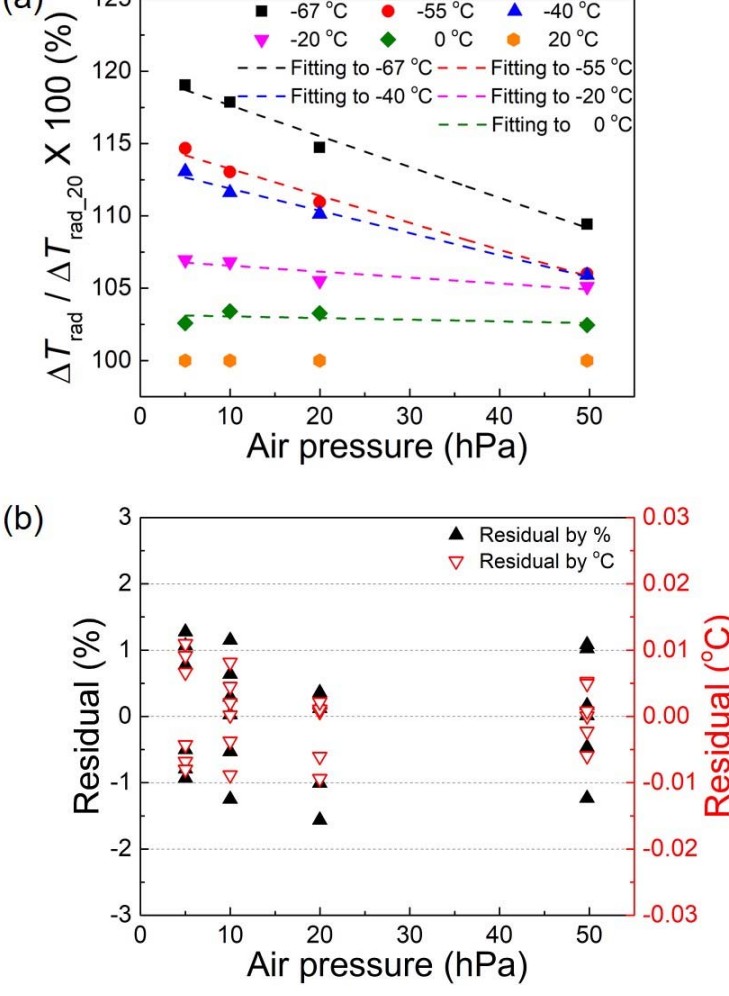

**Figure 4**. (a) Effect of temperature on $\Delta T_{rad}$ normalized by that at 20 °C ($\Delta T_{rad\_20}$ = 100 %) and (b) residuals of linear fittings as a function of the air pressure.

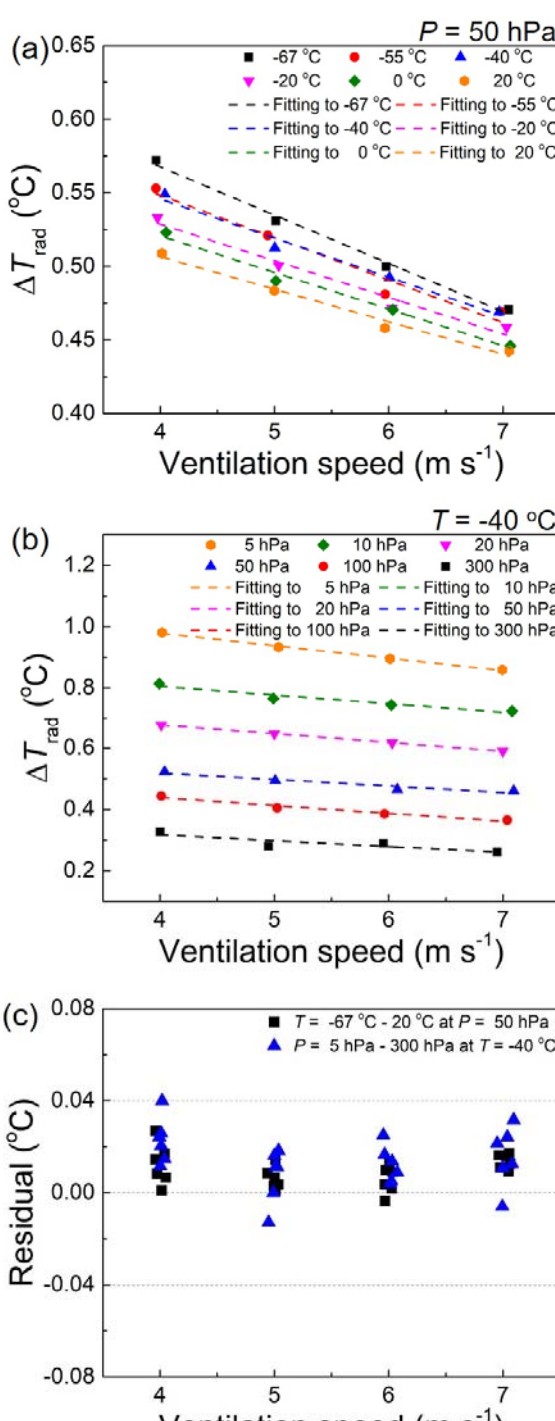

**Figure 5.** Effect of ventilation speed on $\triangle T_{rad}$ at (a) $P = 50$ hPa at different temperatures and (b) $T = -40$ °C at different air pressure values. (c) Residuals as a function of the ventilation speed when Eq. (9) is used.

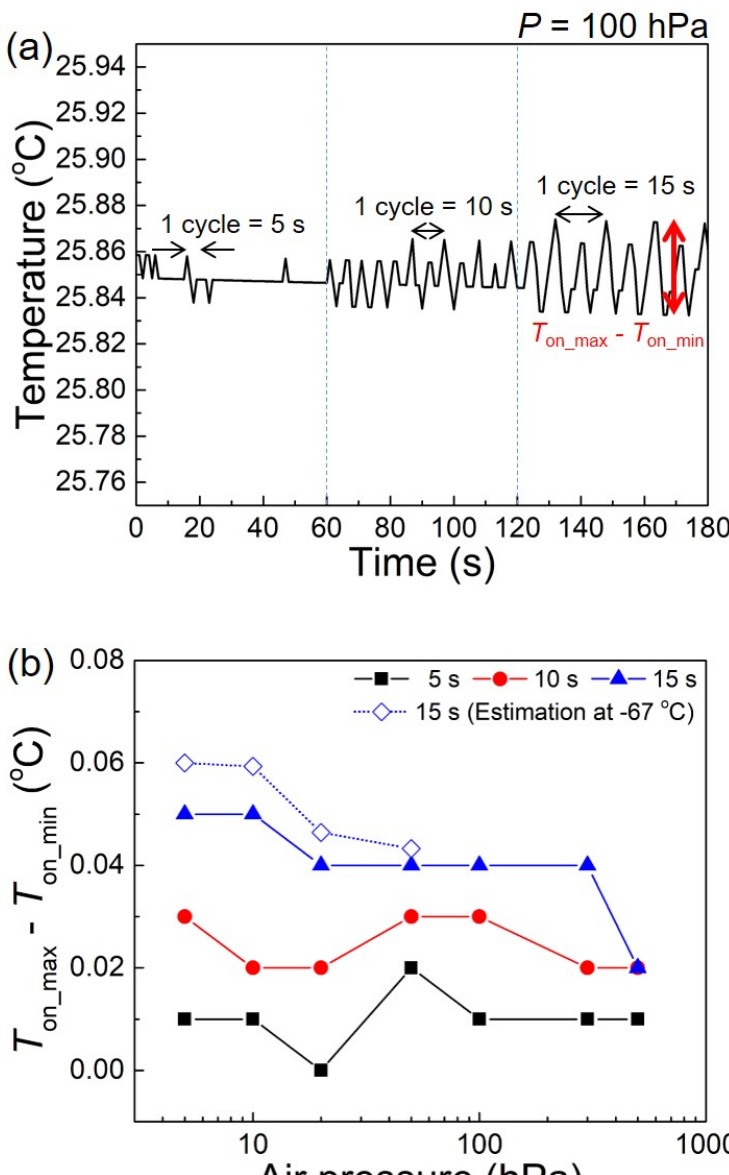

**Figure 6.** (a) Effect of sensor rotation with varied cycles (5 s, 10 s, and 15 s) at $T = 25\ °C$ and $v = 5\ \text{m·s}^{-1}$ and (b) difference in the maximum and minimum temperature values ($T_{\text{on\_max}} - T_{\text{on\_min}}$) as a function of the air pressure. $T_{\text{on\_max}} - T_{\text{on\_min}}$ at 100 hPa and 5 hPa at –67 °C are estimated using Eqs. (6), (7), and (8).

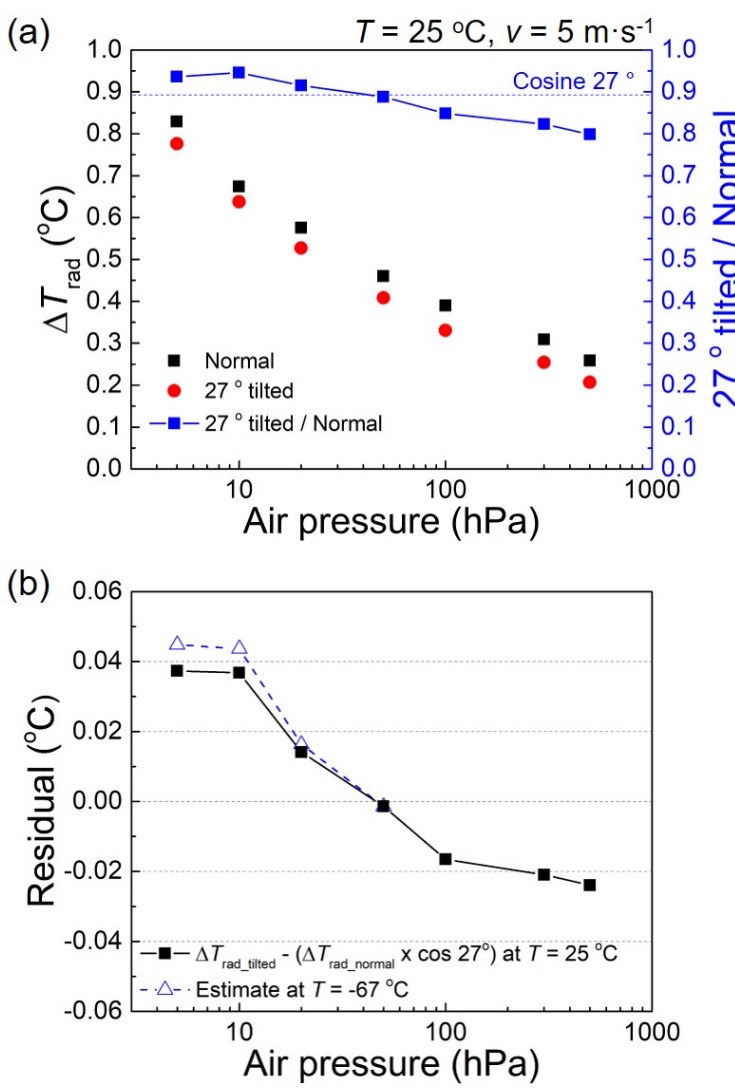

**Figure 7**. (a) Effect of tilting of the sensor boom showing (left y-axis) $\Delta T_{rad}$ with normal ($\Delta T_{rad\_normal}$) and 27° tilted position ($\Delta T_{rad\_tilted}$) and the ratio between them (right y-axis). (b) Residual between $\Delta T_{rad\_tilted}$ and $\Delta T_{rad\_normal} \times \cos 27°$ at $T = 25$ °C and the estimate of the residual at $T = -67$ °C by using Eqs. (6), (7), and (8).

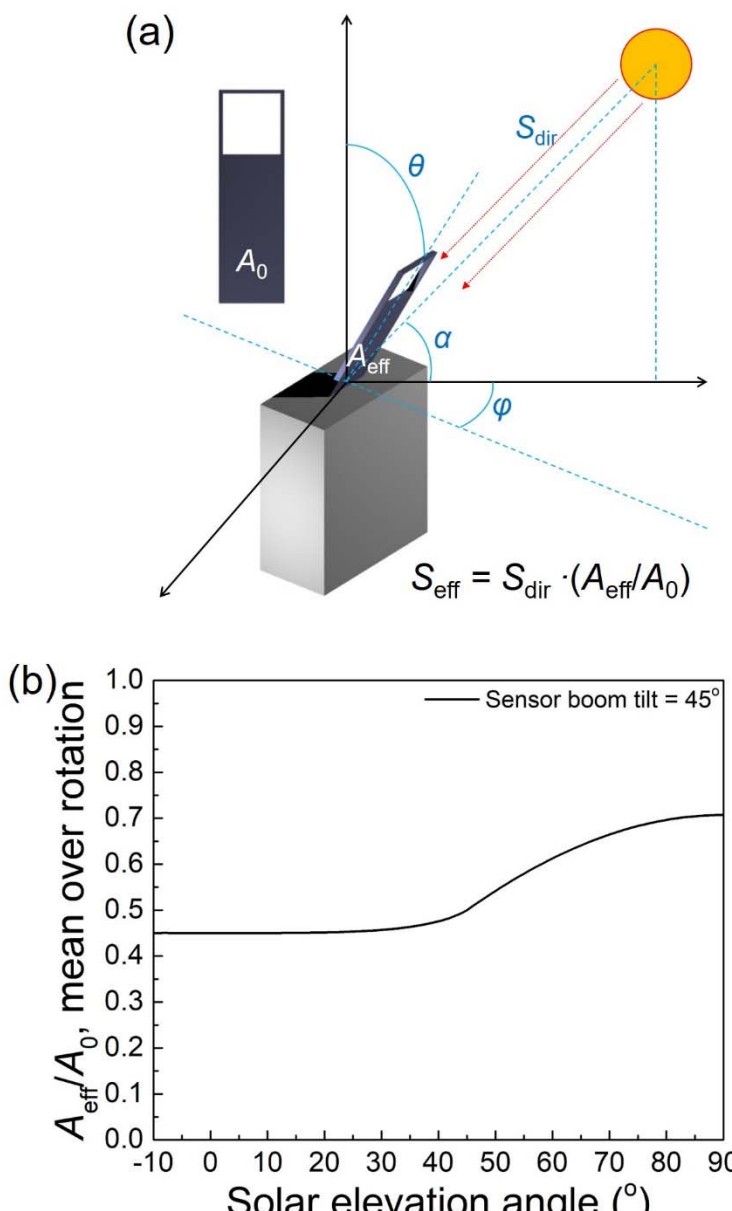

$$S_{eff} = S_{dir} \cdot (A_{eff}/A_0)$$

**Figure 8**. (a) Schematic diagram for the calculation of effective solar irradiance to the sensor. $\theta$, $\alpha$ and $\varphi$ are the tilting angle of the sensor boom, solar elevation angle and azimuthal angle, respectively. (b) Effective irradiation area of the sensor boom obtained by the mean over rotating radiosonde ($\varphi$) as a function of solar elevation angle ($\alpha$) with the tilting anlge $\theta = 45°$.

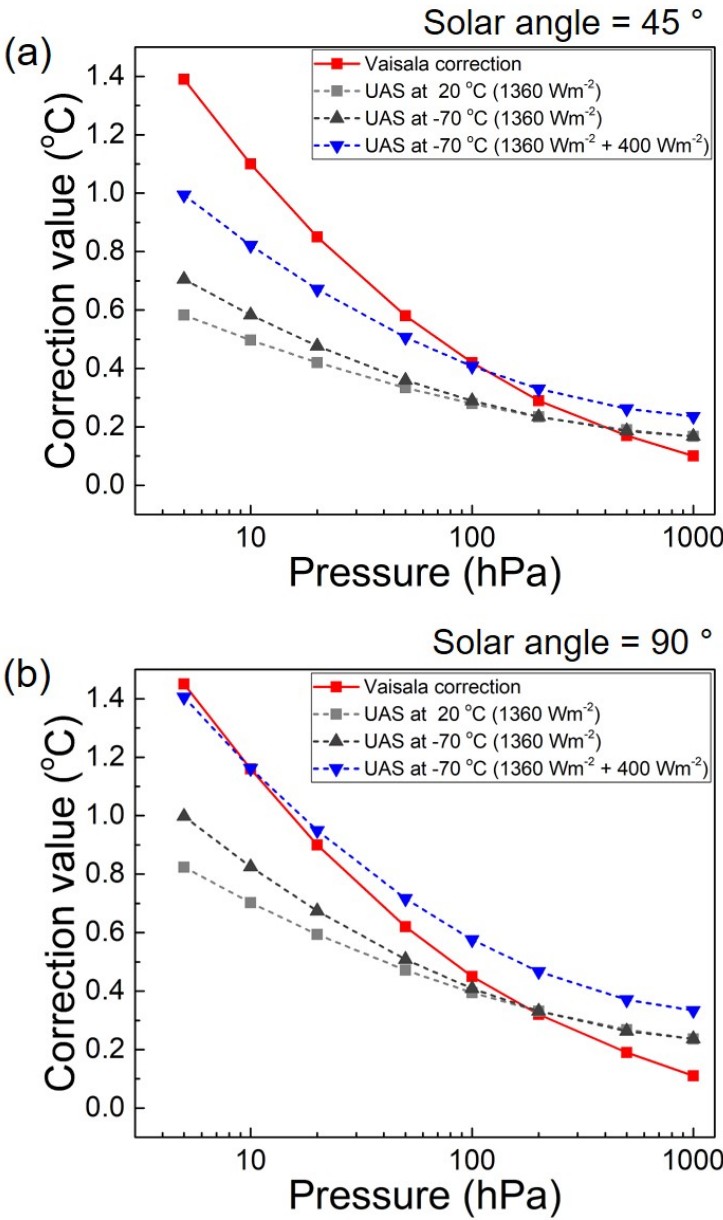

**Figure 9**. Comparison of the radiation correction value between the Vaisala and the UAS as a function of pressure when the solar elevation angle is (a) $\alpha = 45°$ and (b) $\alpha = 90°$ with a boom tilting angle of $\theta = 45°$.