# Peer review of "Radiation correction and uncertainty evaluation of RS41 temperature sensors by using an upper-air simulator"

_Atmospheric Measurement Techniques, 2021_

## Referee Comment (RC1)

Review of Radiation correction and uncertainty evaluation of RS41 temperature sensors by using an upper-air simulator
Lee et al.
AMT-2021-246

The paper describes the application of the in house developed upper air simulator (UAS) to investigate the radiation temperature error of the Vaisala RS41 radiosonde.
The UAS is used to simulate the conditions, both radiation and ambient temperature, that are encountered in the upper atmosphere during a radiosounding.
The main finding of the paper is that lowering the air temperature in the UAS from +20 to -67C leads to an increase of approximately 20% of the radiation temperature error, which the authors postulate is due to changes in the efficiency of the -radiative and conductive- energy transfer mechanisms.
This work is very relevant, because it reports on a manufacturer-independent and traceable method to characterize and quantify error sources in radiosonde temperature measurements. This is an essential prerequisite for providing reference quality data that is suitable for e.g. climate monitoring, especially since the corrections applied by Vaisala in the processing of their radiosonde data are not disclosed to the user.
However, several improvements to the manuscript are necessary to make it suitable for publication.

**Major comments**

My overall impression is that the manuscript is lacking in interpretation of the findings. The method and the results are presented all right, but there is no real attempt made to come to a deeper understanding of the results.
The main message of the paper is the essential role of the ambient  temperature during the measurements, and that lowering the air temperature from +20 to -67C leads to an increase of approximately 20% of the observed radiation temperature error. This is a considerable effect, with potentially far-reaching consequences for the uncertainty estimates of radiosonde data products. Therefore this observed temperature effect requires more elaborate discussion than that is currently included in the manuscript, and the authors should make an attempt to provide a quantifiable physical justification and/or explanation for this observed temperature-dependent difference.

There should be a clear separation between the description of the measurements and the presentation of the results. Currently these are entangled in section 3. I recommend presenting the description and the results in different sections.
Furthermore, provided a more extensive discussion of the assumptions that are made in the data analysis, together with their effect on the quality or uncertainty of the correction.

With regard to the representativeness of the conditions in the UAS compared to the real atmosphere: there are significant differences between the long wave radiation environment in the laboratory setup and in the free atmosphere. In the laboratory setup there is a uniform background emitted by the walls of the measurement chamber, with only a small temperature difference (approximately 1-2K) with respect to the temperature sensor. In the free atmosphere the long-wave radiation background is composed of, amongst others, contributions from the air masses and surface below and from the cold cosmic background

above. Therefore, the question is how well the environment inside the UAS is representative for the conditions encountered in the stratosphere, and it cannot be excluded that the observed low-temperature effect is to some extent specific for the conditions inside the measurement chamber.
I realize that it is a challenge to recreate the exact atmospheric and radiative conditions of the upper atmosphere in a laboratory set up, but the authors should discuss this.
In connection to this it would be very useful to use the dual thermistor that is discussed in another paper by the same authors (Lee et al. Meteorol. Appl. 25: 283–291 (2018), doi: 10.1002/met.1690) to make a connection between tests in the UAS and in situ measurements at high altitude. This should provide an answer to the question whether measurements in the UAS can replicate the temperature difference between both sensors of the dual thermistor radiosonde at high altitude for an ambient temperature of -67C?

The introduction should include more discussion of, and references to, other relevant work.

The influence of the thermal coupling between sensor boom and temperature sensor for the radiation error should be discussed in more detail.

The differences with the Vaisala radiation correction are listed in Table 11, but without much further discussion or interpretation of these differences. The UAS-based findings should be presented in a plot, together with Vaisala correction, so that it is easier for the reader to see the differences between both correction methods. At this place a discussion of the observed altitude-dependence of these differences would be appropriate.

In the description of the experiments the following information appears to be missing:
the length of the exposure times, and the number of repetitions.

Show uncertainties in the plots of figures 2-6

Please provide a more detailed description of how Delta_T is determined. Is this the difference between the radiosonde temperature and the air temperature in the UAS?

**Specific comments**

l23-24: it should be mentioned that radiosondes perform in situ measurements
l26: add pressure to the list of ECVs
l26: (concerning the high accuracy), provide an estimate how high this accuracy is.
l28: This sentence needs to be rephrased. I assume you want to say something like...
l29-30: rephrase: may lead to inhomogeneities in data records due to the use of different radiosonde models.
l33: of the ECVs -> of selected ECVs
l37: rephrase: are exposed to solar radiation, which leads to radiative heating of the temperature sensor.
l37: As background info to the reader, provide a typical estimate of the radiation temperature error for current (and previous) radiosondes.
l40: A proper reference for the need to reduce sensor size would be de Podesta et al 2020.

l41: This sentence is unclear, please rephrase. What do you mean by the combined effect of solar radiation?

l48: Add a one-line description of how Schmidlin1986 corrected the radiation error.

l50-52: include the quantitative result of Philipona2013, i.e. the magnitude of the correction they derived

l57: add proper reference (von Rohden et al. AMT 2021, doi:10.5194/amt-2021-187)

l59-64: The chamber system used by Lee2018, Lee2020, and in this manuscript is one and the same. This should be clear from the text.

l65-66: Add a sentence explaining how the radiation correction is evaluated. This is discussed in detail in the later sections, but a brief mentioning of the principle should be given in the introduction.

l66: Provide additional details on the UAS. The possibility to tilt and rotate the radiosonde are important innovations that should be mentioned here.

l81-82: do you mean the temperature of the test chamber, or the temperature of the air inside the test chamber?

l92: What are the results of the laser Doppler velocimetry characterisation of the airflow inside the test chamber?

l96: It is not clear whether 980 W/m2 is the flux at the position of the radiosonde's sensor, so inside the test chamber, or if it denotes the flux at the window of the test chamber.

l102-103: Should the study still be seen as proof of concept?

l100: mention the accuracy at which the pyranometer is calibrated at KRISS.

l107: Remove 'it seems that'. The following sentence makes clear that you know the manufacturer's radiation correction is not applied.

l116: considered -> employed

l119: Change section number to 3

l128: I don't understand this sentence. Do you mean that the relative uncertainty of Delta T is larger at high pressure because of its smaller value?

l129: Motivation for using exponential functions for fitting?

l144-146: See remark in the major comments above about physical justification for the different radiation error at low temperature. Changing the temperature from +20C to -67C has a 20% effect on the observed radiation error. This is a significant difference which merits a thorough discussion of the effects involved. Preferably with appropriate references. For example, what could be the effect of temperature to the convective heat exchange?

l170-171: This is a bold conclusion that should be reached only after thorough verification and discussion.

l177: Despite the limited investigated range of ventilation, is it reasonable to assume a simple linear relationship between radiation effect and ventilation speed?

l205-206: please rephrase. Say something along the lines that the exposed surface of the sensor boom depends on the incidence angle, and passes through a maximum twice during a full rotation.

Table 10: Vaisala provides more detailed information on calibration uncertainty of the temperature sensor in the White Paper 'Vaisala Radiosonde RS41 Measurement Performance', B211356EN-B-LOW-v3

Figure 5: the plots show the effect of the rotation on the sensorboom on the radiative heating of the temperature sensor. Figure 11 of von Rohden 2021 presents the same effect, however

with a considerably larger magnitude (up to 0.3K for p=7hPa and 16s rotation period). This difference in findings should be addressed by the authors.

The amplitude and shape of the oscillations also depend considerably on the incidence angle (equivalent to solar elevation), which is not investigated in the study. This may lead to considerably larger $T_{(on\_max)} - T_{(on\_min)}$ in Fig. 5b, and therefore to higher uncertainty estimates due to sonde rotation.

Fig 5 a) shows the results for 100 hPa, where the effect is rather small. It would be more meaningful showing the results for p=5 hPa.

The bump around 100 hPa in the pressure dependence of the amplitude for the black and red curve is striking. Does this reflect the uncertainty of the measurement?

Finally, mention in the caption the values of v, and T for the data in the plot.

---

## Author Comment (AC1)

**Referee #1**

**General Comment:** The paper describes the application of the in house developed upper air simulator (UAS) to investigate the radiation temperature error of the Vaisala RS41 radiosonde. The UAS is used to simulate the conditions, both radiation and ambient temperature, that are encountered in the upper atmosphere during a radiosounding. The main finding of the paper is that lowering the air temperature in the UAS from +20 to -67C leads to an increase of approximately 20% of the radiation temperature error, which the authors postulate is due to changes in the efficiency of the -radiative and conductive- energy transfer mechanisms. This work is very relevant, because it reports on a manufacturer-independent and traceable method to characterize and quantify error sources in radiosonde temperature measurements. This is an essential prerequisite for providing reference quality data that is suitable for e.g. climate monitoring, especially since the corrections applied by Vaisala in the processing of their radiosonde data are not disclosed to the user. However, several improvements to the manuscript are necessary to make it suitable for publication.

→ We thank the Reviewer for valuable comments. We have revised the manuscript to comply with the Reviewer's comment as below.

**Major comments:** My overall impression is that the manuscript is lacking in interpretation of the findings. The method and the results are presented all right, but there is no real attempt made to come to a deeper understanding of the results. The main message of the paper is the essential role of the ambient temperature during the measurements, and that lowering the air temperature from +20 to -67C leads to an increase of approximately 20% of the observed radiation temperature error. This is a considerable effect, with potentially far-reaching consequences for the uncertainty estimates of radiosonde data products. Therefore this observed temperature effect requires more elaborate discussion than that is currently included in the manuscript, and the authors should make an attempt to provide a quantifiable physical justification and/or explanation for this observed temperature-dependent difference.

→ We have added theoretical discussion on the temperature effect in Section 3.2 with **Figure 3** and **Table 2**.

[revised manuscript text omitted]

There should be a clear separation between the description of the measurements and the presentation of the results. Currently these are entangled in section 3. I recommend presenting the description and the results in different sections. Furthermore, provided a more extensive discussion of the assumptions that are made in the data analysis, together with their effect on the quality or uncertainty of the correction.

→ We prefer the current format of the manuscript to explain the effect of each parameter on the radiation correction one by one and show how the correction formula is updated to incorporate each effect.

With regard to the representativeness of the conditions in the UAS compared to the real atmosphere: there are significant differences between the long wave radiation environment in the laboratory setup and in the free atmosphere. In the laboratory setup there is a uniform background emitted by the walls of the measurement chamber, with only a small temperature difference (approximately 1-2K) with respect to the temperature sensor. In the free atmosphere the long-wave radiation background is composed of, amongst others, contributions from the air masses and surface below and from the cold cosmic background above. Therefore, the question is how well the environment inside the UAS is representative for the conditions encountered in the stratosphere, and it cannot be excluded that the observed low-temperature effect is to some extent specific for the conditions inside the measurement chamber. I realize that it is a challenge to recreate the exact atmospheric and radiative conditions of the upper atmosphere in a laboratory set up, but the authors should discuss this. In connection to this it would be very useful to use the dual thermistor that is discussed in another paper by the same authors (Lee et al. Meteorol. Appl. 25: 283–291 (2018), doi: 10.1002/met.1690) to make a connection between tests in the UAS and in situ measurements at high altitude. This should provide an answer to the question whether measurements in the UAS can replicate the temperature difference between both sensors of the dual thermistor radiosonde at high altitude for an ambient temperature of -67C?

→ We have learned that the net longwave radiative cooling from the sensor is negligible ($\sim 10^{-6}$ W) compared to the convective cooling (200 W) in the above calculation. According to the calculation, the observed temperature effect on $\Delta T_{rad}$ can be explained by the change of air properties at low temperatures which is not related with the use of a measurement chamber.

For your information, dual thermistor radiosonde (DTR) was also tested using the UAS and the result was submitted to the same Journal (ID: amt-2021-343).

The introduction should include more discussion of, and references to, other relevant work.

→ Throughout revisions to comply with the specific comments of the Reviewer, the Introduction is reinforced with discussions of relevant works.

The influence of the thermal coupling between sensor boom and temperature sensor for the radiation error should be discussed in more detail.

→ The thermal coupling between sensor boom and temperature sensor is discussed.

**Added Statement (Line 205-212)**: It was previously observed that the temperature rise of RS92 was initially fast due to the small thermal mass of the sensor and subsequently slow (Dirksen et al., 2014). More recently, the temperature of RS41 oscillated when the radiosonde was rotating under irradiation (Von Rohden et al., in review, 2021). These observations are attributed that the heating of the sensor boom with comparably large area is coupled to the heating of the temperature sensor. Since the conductive heat transfer from the sensor boom is missing in the above theoretical calculation, the comparison in Fig. 3(b) may show the effect of the sensor boom on $\Delta T_{rad}$. Interestingly, the growth of $\Delta T_{rad}$ of the theoretical calculation is less steep than that of the experiment as the pressure is decreased to 5 hPa. This may imply that the heat transfer from the sensor boom becomes significant especially at low pressures.

The differences with the Vaisala radiation correction are listed in Table 11, but without much further discussion or interpretation of these differences. The UAS-based findings should be presented in a plot, together with Vaisala correction, so that it is easier for the reader to see the differences between both correction methods. At this place a discussion of the observed altitude-dependence of these differences would be appropriate.

→ As suggested, the radiation corrections by the UAS and Vaisala are presented together by plots as shown in Fig.9. To explain the situation, Figure 8 is also added.

**Before:** The radiation correction of RS41 by the UAS is based on Eqs. (13) and (14) for different pressure ranges. Although the conditions for the UAS correction are different from those considered by the manufacturer, a rough comparison of the radiation corrections is presented in Table 11. For the UAS correction, the solar irradiance is assumed to be S = 1360

W·m$^{-2}$ at all pressure values. Depending on the effective irradiance ($S_{eff}$), the UAS correction value should be revised in a proportional manner using Eqs. (13) and (14).

**After (Line 397-413)**: In order to apply the correction formula to actual soundings, the effective irradiance to the sensor should be known. However, radiosondes constantly change positions with respect to the solar irradiation through rotation and pendulum motion, the calculation of effective irradiance resorts on the mean of effective irradiance over the motion of radiosondes. Figure 8(a) shows a schematic diagram of a radiosonde with parameters that affect the effective irradiance $S_{eff}$ on the sensor. Then, the effective irradiance to the sensor can be calculated as follows:

$$S_{eff} = S_{dir} \cdot \left| \cos\alpha \, \cos\Theta \, \cos\varphi - \sin\Theta \, \sin\alpha \right|, \tag{18}$$

$S_{dir}$ is solar direct irradiance, $\theta$ is boom tilting angle, $\alpha$ is solar elevation angle and $\varphi$ is azimuthal angle. The effective irradiation area ($A_{eff}/A_0$) on the sensor boom is averaged over rotation ($\varphi$) with a fixed tilting angle $\theta = 45\,°$ and plotted as a function of the solar elevation angle as shown in Fig. 8(b). Using this effective irradiance, the radiation correction by the UAS is obtained and compared with that of the manufacturer at two different $\alpha$ (45 ° and 90 °) as shown in Fig. 9. For the UAS correction, the solar direct irradiance is assumed to be 1360 W·m$^{-2}$ at all pressure values. To simulate the albedo effect, the radiation correction with additional irradiance of 400 W·m$^{-2}$ is also calculated. Consequently, the radiation correction of the UAS is smaller than the Vaisala by about 0.5−0.7 °C at −70 °C and 5 hPa when only the solar direct irradiance (1360 W·m$^{-2}$) is considered with the solar elevation angle $\alpha = 45−90\,°$. When the albedo effect is additionally included (400 W·m$^{-2}$), the gap between the two corrections is reduced to 0.04−0.4 °C at −70 °C and 5 hPa with $\alpha = 45−90\,°$. The radiation corrections of the manufacturer and the UAS at some representative conditions are summarized in **Table 11**.

**Modified Table (Table 11): Table 11** is modified to include the radiation correction of the UAS obtained by the above method.

In the description of the experiments the following information appears to be missing: the length of the exposure times, and the number of repetitions.

→ The length of the exposure time and the number of repetitions are added.

**Added Statement (Line 148)**: The duration of irradiation is 120 s and the measurement is repeated three times.

Show uncertainties in the plots of figures 2-6

→ The data in Figure 2 represents the mean and the standard deviation of three repeated measurements on a single RS41 unit. Since standard deviations are small (max. 0.014 °C), they are not seen clearly. Therefore, the biggest standard deviation is mentioned in the manuscript.

Standard deviations are newly added to Figs. 3 and 4, but not to Figs. 5, 6, and 7 because these experiments were conducted once.

**Added Statement (Line 152-153)**: The data represents the mean and the standard deviation of three repeated measurements on a single RS41 unit. The biggest standard deviation was 0.014 °C.

Please provide a more detailed description of how Delta_T is determined. Is this the difference between the radiosonde temperature and the air temperature in the UAS?

→ It is mentioned in the manuscript.

**Before:** The temperature rise due to irradiation ($\Delta T_{rad}$) is defined as the difference in the temperatures with irradiation ($T_{on}$) and without irradiation ($T_{off}$); $\Delta T_{rad} = T_{on} - T_{off}$.

**After (Line 146-148)**:    The temperature rise due to irradiation ($\Delta T_{rad}$) is defined as the difference in the temperatures with irradiation ($T_{on}$) and without irradiation ($T_{off}$) as previously reported (Lee et al., 2020); $\Delta T_{rad} = T_{on} - T_{off}$.

**Specific comments**

**l23-24:** it should be mentioned that radiosondes perform in situ measurements

→ The sentence is changed to include "in situ measurement".

**Before:** Radiosondes are telemetry devices that include various sensors to measure data that are transmitted to a ground receiver while the device is carried by a weather balloon to an altitude of approximately 35 km.

**After (Line 25-27)**: Radiosondes are telemetry devices that include various sensors to perform in situ measurements and transmit the measured data to a ground receiver while the device is carried by a weather balloon to an altitude of approximately 35 km.

**l26:** add pressure to the list of ECVs

→ Pressure is added to the list of ECVs.

**Before:** essential climate variables (ECVs) such as the temperature, water vapour, wind speed, and wind direction

**After (Line 28)**: essential climate variables (ECVs) such as the temperature, water vapour, pressure, wind speed, and wind direction

**l26:** (concerning the high accuracy), provide an estimate how high this accuracy is.

→ The claimed accuracy of Vaisala RS41 is presented.

**Before:** Owing to their high accuracy

**After (Line 29):** Owing to their high accuracy of 0.3 to 0.4 K claimed by manufacturers

**l28:** This sentence needs to be rephrased. I assume you want to say something like...

→ The sentence is changed.

**Before:** Notably, an effective method to evaluate the measurement accuracy specified by manufacturers remains to be specified.

**After (Line 30-33):** However, evaluation methods for their sensor accuracy are not fully disclosed to users. Operation principle of laboratory setups, algorithms to correct measurement errors, and corresponding uncertainty evaluations are prerequisites for a reference data product.

**l29-30:** rephrase: may lead to inhomogeneities in data records due to the use of different radiosonde models.

→ Rephrased.

**Before:** may lead to inhomogeneity among users, including upper-air observatories that use different radiosonde models.

**After (Line 33-34):** may lead to inhomogeneities in data records due to the use of different radiosonde models.

**l33:** of the ECVs -> of selected ECVs

→ Done.

**Before:** of the ECVs

**After (Line 37):** of selected ECVs

**l37:** rephrase: are exposed to solar radiation, which leads to radiative heating of the temperature sensor.

→ Rephrased.

**Before:** are exposed to solar radiation, and the radiative heating increases the temperature to more than the air temperature

**After (Line 41-42)**: are exposed to solar radiation, which leads to radiative heating of the temperature sensor.

l37: As background info to the reader, provide a typical estimate of the radiation temperature error for current (and previous) radiosondes.

→ Typical estimates of the radiation correction are presented.

**Added Sentence (Line 42-45):** According to the last intercomparison of high quality radiosonde systems (Nash et al., 2011), radiation correction values applied by manufacturers were distributed from 0.6 to 2.3 K at 10 hPa. More recently, according to the radiation correction of Vaisala RS41, it is increased from 0.83 to 1.16 K at 10 hPa as solar angle is elevated from 10 ° to 90 °.

l40: A proper reference for the need to reduce sensor size would be de Podesta et al 2020.

→ The reference is added.

**Added Reference (Line 48):** de Podesta *et al*. 2018 "Air temperature sensors: dependence of radiative errors on sensor diameter in precision metrology and meteorology" is added.

l41: This sentence is unclear, please rephrase. What do you mean by the combined effect of solar radiation?

→ The sentence is modified to deliver the meaning more clearly.

**Before:** However, it is not possible to eliminate the combined effect of solar irradiation

**After (Line 49-50)**: Nevertheless, the effect of solar irradiation cannot be eliminated and thus should be corrected properly.

l48: Add a one-line description of how Schmidlin1986 corrected the radiation error.

→ A short description on Schmidlin *et al*. 1986 is added.

**Added Sentence (Line 57-59):** Radiation correction was estimated by using radiosondes equipped with four thermistors having coatings with different spectral responses, i.e., emissivities and absorptivities (Schmidlin et al., 1986).

**l50-52:** include the quantitative result of Philipona 2013, i.e. the magnitude of the correction they derived.

→ A quantitative result of Philipona *et al*. 2013 is added.

**Added Sentence (Line 62-63):** As a result, radiation correction was obtained by a linear function of geopotential height which gives 1 K at 32 km.

**l57**: add proper reference (von Rohden et al. AMT 2021, doi:10.5194/amt-2021-187)

→ The reference is added properly.

**Added Reference:** von Rohden, C., Sommer, M., Naebert, T., Motuz, V., and Dirksen, R. J.: Laboratory characterisation of the radiation temperature error of radiosondes and its application to the GRUAN data processing for the Vaisala RS41, Atmos. Meas. Tech. Discuss. [preprint], https://doi.org/10.5194/amt-2021-187, in review, 2021.

**l59-64:** The chamber system used by Lee2018, Lee2020, and in this manuscript is one and the same. This should be clear from the text.

→ Two references are separated.

**Before:** Notably, the existing studies based on other chamber systems reported that the solar-irradiation-induced temperature rise of sensors increases as the air temperature is decreased (Lee et al., 2018a; Lee et al., 2020).

**After (Line 70-71)**: Notably, a previous study based on a chamber system reported that the solar-irradiation-induced temperature rise of sensors increases as the air temperature is decreased (Lee et al., 2018a).

**Before:** In this study, the uncertainty in the radiation correction of a Vaisala RS41 temperature sensor is evaluated using the UAS at KRISS.

**After (Line 76-77)**: In this study, the uncertainty in the radiation correction of a Vaisala RS41 temperature sensor is evaluated using the UAS developed at KRISS (Lee et al., 2020).

**l65-66:** Add a sentence explaining how the radiation correction is evaluated. This is discussed in detail in the later sections, but a brief mentioning of the principle should be given in the introduction.

→ A sentence explaining the evaluation on the radiation correction is added.

**Added Sentence (Line 77-79):** It is shown how the uncertainty of each environmental parameter and radiosonde movements in the UAS contributes to the uncertainty of RS41 through a radiation correction formula obtained by a series of radiation experiments.

**l66:** Provide additional details on the UAS. The possibility to tilt and rotate the radiosonde are important innovations that should be mentioned here.

→ The possibility to tilt and rotate the radiosonde is mentioned.

**Before:** along with the addition of new functions to consider the effect of the rotation and tilting of the sensor.

**After (Line 79-80):** along with the addition of new functions to consider the effect of the rotation and tilting of the sensor that are an important progress from the previous version of the UAS.

**l81-82:** do you mean the temperature of the test chamber, or the temperature of the air inside the test chamber?

→ The climate chamber is used to control the temperature of both test chamber and air.

**Before:** The test chamber is inside a climate chamber (Tenney environmental, Model: C64RC) to control the temperature. The working space of the climate chamber is sized 1219 mm × 1219 mm × 1219 mm.

**After (Line 91-97):** The test chamber is inside a climate chamber (Tenney environmental, Model: C64RC) of which working space is 1219 mm × 1219 mm × 1219 mm. The temperature of the test chamber is controlled by the climate chamber. Air is precooled before entering into the climate chamber by passing through a heat exchanger in a separate bath (Kambic metrology, Model: OB-50/2 ULT) of which temperature is lower than that of the climate chamber by about 5 °C. The temperature of the precooled air is then adjusted to that of the climate chamber while passing through the second heat exchanger (9.3 m in length) in the climate chamber before entering into the test chamber.

**l92:** What are the results of the laser Doppler velocimetry characterisation of the airflow inside the test chamber?

→ Description on the measurement by the laser Doppler velocimetry inside the test chamber is added.

**Added sentence (Line 103-104):** Thus, the reference value and SI traceability of the ventilation speed are obtained by using the sonic nozzles in the UAS.

**Before:** The generated air flow is measured through laser Doppler velocimetry to investigate the spatial gradient in the test chamber.

**After (Line 108-111):** The generated air flow is measured through laser Doppler velocimetry (LDV) (Dantec, Model: BSA F60) to investigate the spatial gradient in the test chamber. Ar-ion laser (3W) having a wavelength of 514.5 nm is used with a focal length of 400.1 mm and nominal beam spacing of 33 mm.

**Before:** The spatial gradient of the ventilation speed in the test chamber is measured through laser Doppler velocimetry at KRISS.

**After (Line 327-332):** The spatial gradient of the ventilation speed in the test chamber is measured through the LDV at KRISS. The measurement dimension using the LDV was 35 mm x 35 mm around the sensor (central) location with 5 mm interval (49 points) in the test chamber (50 mm x 50 mm). The measurement was performed at the condition of $v$ = 4.67 m·s$^{-1}$ (reference value), $P$ = 550 hPa, and room temperature. The LDV value averaged over the measurement area (35 mm x 35 mm) was 4.63 m·s$^{-1}$. The difference between the reference and the measurement average is assumed to have a rectangular probability distribution for the calculation of the uncertainty of spatial gradient.

**l96:** It is not clear whether 980 W/m2 is the flux at the position of the radiosonde's sensor, so inside the test chamber, or if it denotes the flux at the window of the test chamber.

→ The flux at the position of the radiosonde sensors inside the test chamber is 980 W/m$^2$.

**Before:** A constant irradiance of 980 W·m$^{-2}$ is adopted throughout this study.

**After (Line 115-116):** A constant irradiance of 980 W·m$^{-2}$ at the position of the radiosonde sensors inside the test chamber is adopted throughout this study.

**l102-103:** Should the study still be seen as proof of concept?

→ The phrase "As a proof of concept" is removed.

**Removed phrase:** As a proof of concept

**l100:** mention the accuracy at which the pyranometer is calibrated at KRISS.

→ The calibration uncertainty is mentioned.

**Before:** The pyranometer is calibrated at KRISS.

**After (Line 119-120):** The pyranometer is calibrated at KRISS and the uncertainty is 1% of the measured value with a coverage factor $k = 1$.

**l107:** Remove 'it seems that'. The following sentence makes clear that you know the manufacturer's radiation correction is not applied.

→ The phrase is removed as suggested.

**Removed phrase:** It seems that

**l116:** considered -> employed

→ The word is changed as suggested.

**Before:** considered

**After (Line 138):** employed

**l119:** Change section number to 3

→ Done.

**Before:** 2 Experiment Details

**After (Line 141):** 3 Experiment Details

**l128:** I don't understand this sentence. Do you mean that the relative uncertainty of Delta T is larger at high pressure because of its smaller value?

→ The relatively larger uncertainties at high pressures indicates that $U(\Delta T_{rad})/ \Delta T_{rad}$ becomes larger because $\Delta T_{rad}$ is decreased at high pressures. Thus, the temperature effect is not clearly observable at high pressures.

**Before:** This phenomenon can be attributed to the relatively larger uncertainties in $\Delta T_{rad}$ at high pressures in the UAS.

**After (Line 156-157):** This phenomenon can be attributed that the uncertainty of $\Delta T_{rad}$ becomes relatively larger with respect to $\Delta T_{rad}$ as $\Delta T_{rad}$ is decreased at high pressures in the UAS.

**l129:** Motivation for using exponential functions for fitting?

→ Exponential fittings are replaced by polynomial fittings with $\log_{10} P$ to provide a single fit over entire pressure range of 5−500 hPa (Reviewer 2's suggestion). Consequently, residuals by polynomial fittings (±0.03 °C) are smaller than those by exponential fittings (±0.04 °C) as shown in Fig. 2(b). Therefore, polynomial fittings are newly used to obtain radiation correction formula throughout the revised manuscript.

**Modified Figure (Figure 2):** Figure 2(a) is replotted by using parabola fittings of $\log_{10} P$ and the residual in Fig. 2(b) is newly obtained accordingly.

**Modified Equation (Equation 1, 9, 10 & 11):** The backbone of the Equations is changed to $\Delta T_{\mathrm{rad}} = A_0(T) + B_0(T) \cdot \log(P) + C_0(T) \cdot [\log(P)]^2$

**Modified Table (Table 1):** The original **Table 1** is removed and **Table 2** is relabeled to **Table 1. Table 1** is modified to include information on new coefficients of $A_0(T)$, $B_0(T)$, and $C_0(T)$.

**l144-146:** See remark in the major comments above about physical justification for the different radiation error at low temperature. Changing the temperature from +20C to -67C has a 20% effect on the observed radiation error. This is a significant difference which merits a thorough discussion of the effects involved. Preferably with appropriate references. For example, what could be the effect of temperature to the convective heat exchange?

→ As suggested, the theoretical discussion on the effect of temperature in terms of convective heat exchange is added in the revised manuscript.

**l170-171:** This is a bold conclusion that should be reached only after thorough verification and discussion.

→ Both experimental and theoretical works indicate that the radiation correction is dependent on temperature because the properties of air change with temperature. We do not think that the statement is a bold conclusion.

**l177:** Despite the limited investigated range of ventilation, is it reasonable to assume a simple linear relationship between radiation effect and ventilation speed?

→ Even though the relationship is not perfectly linear, residuals by using a linear function is not significant. The validity of the linear relationship is newly described in the revised manuscript.

**Added Statement (Line 249-250):** The linear relationship between the ventilation speed and the radiation correction in Eq. (9) is only valid in the range of $4-7$ m·s$^{-1}$. When $v$ is higher than $7$ m·s$^{-1}$ or lower than $4$ m·s$^{-1}$, the formula underestimates the correction value.

**l205-206:** please rephrase. Say something along the lines that the exposed surface of the sensor boom depends on the incidence angle, and passes through a maximum twice during a full rotation.

→ The sentence is changed as suggested.

**Before:** This phenomenon occurs because the sensor boom undergoes similar processes in the first 180° and remaining 180° in a 360° rotation.

**After (Line 269-270):** The exposed surface of the sensor boom depends on the incidence angle, and passes through a maximum twice during a full rotation.

**Table 10:** Vaisala provides more detailed information on calibration uncertainty of the temperature sensor in the White Paper 'Vaisala Radiosonde RS41 Measurement Performance', B211356EN-B-LOW-v3

→ Since $T_{cor} = T_{raw} - \Delta T_{rad}$ in Eq. (16), the uncertainty of $T_{cor}$ requires that of $T_{raw}$ and $\Delta T_{rad}$. $U(\Delta T_{rad})$ is obtained by the UAS experiment. $U(T_{raw})$ is adopted from the White Paper which describes the uncertainty of "repeatability in calibration" is 0.1 °C at $k = 2$. Although Vaisala provides other uncertainties related with the storage and soundings (as a manufacturer), this work is limited to the evaluation of RS41 by a laboratory facility. The uncertainty due to soundings should be added to the total uncertainty when the correction formula is applied to soundings. The White Paper is cited in **Table 10**.

**Added Statement (Line 390-392):** Since Vaisala provides additional uncertainty of reproducibility in sounding (0.15 °C when $P > 100$ hPa, 0.3 °C when $P < 100$ hPa) (Vaisala), this should be added to the total uncertainty of the corrected temperature when the radiation correction formula in Eq. (11) is applied to soundings.

**Figure 5:** the plots show the effect of the rotation on the sensorboom on the radiative heating of the temperature sensor. Figure 11 of von Rohden 2021 presents the same effect, however with a considerably larger magnitude (up to 0.3K for p=7hPa and 16s rotation period). This difference in findings should be addressed by the authors. The amplitude and shape of the oscillations also depend considerably on the incidence angle (equivalent to solar elevation), which is not investigated in the study. This may lead to considerably larger $T_{(on\_max)}$ - $T_{(on\_min)}$ in Fig. 5b, and therefore to higher uncertainty estimates due to sonde rotation. Fig 5 a) shows the results for 100 hPa, where the effect is rather small. It would be more meaningful

showing the results for p=5 hPa. The bump around 100 hPa in the pressure dependence of the amplitude for the black and red curve is striking. Does this reflect the uncertainty of the measurement? Finally, mention in the caption the values of v, and T for the data in the plot.

→ The rotation axis is the temperature sensor itself, not the center of the boom. Therefore, the temperature sensor only spins on the spot and thus the distance between the sensor and the light source does not change at all during the rotation. This implies that the irradiance to the sensor is constant whereas the light incident angle (effective irradiance) to the sensor boom changes with rotation. This explains why the maximum temperature peak appears twice during a single cycle. The fixed rotation axis to the sensor may be the reason why the $T_{on\_max} - T_{on\_min}$ in the UAS is smaller than that of von Rohden *et al.* (0.3 °C). In the work of von Rohden *et al*, although the distance from the light source to the sensor is constant (or slightly changes), that to the sensor boom changes with rotation. This may be the reason why the maximum peak appears once in a single cycle when the sensor boom is close to the light source.

The biggest estimated value of $T_{on\_max} - T_{on\_min}$ (0.06 °C) at $T = -67$ °C and $P = 5$ hPa is used for the uncertainty calculation for all pressures.

The values of *v* and *T* are mentioned in the caption.

**Added Statement (Line 263-265):** The rotation axis is the temperature sensor itself, not the centre of the boom in this work. Therefore, the temperature sensor only spins on the spot and thus the distance between the sensor and the solar simulator does not change during the rotation.

**Added Statement (Line 277-282):** The maximum value of $(T_{on\_max} - T_{on\_min})$ in the UAS (0.05 °C) is much smaller than that of von Rohden *et al.* (0.3 °C) (Von Rohden et al., in review, 2021). In the work of von Rohden *et al*, although the distance from the light source to the sensor is constant, that to the sensor boom changes with rotation. This may be the reason why the maximum peak appears once in a full cycle when the sensor boom is close to the light source and the $(T_{on\_max} - T_{on\_min})$ is bigger than this work. The relatively small $(T_{on\_max} - T_{on\_min})$ with respect to $\Delta T_{rad}$ observed in this work suggests that the contribution of the thermal conduction to $\Delta T_{rad}$ is small compared to that by the direct irradiation of the sensor.

**Before:** (a) Effect of sensor rotation with varied cycles (5 s, 10 s, and 15 s)

**After (Figure 6 Caption):** (a) Effect of sensor rotation with varied cycles (5 s, 10 s, and 15 s) at $T = 25$ °C and $v = 5$ m·s$^{-1}$

---

## Author Comment (AC2)

**Referee #2**

**Summary:**

The manuscript by Lee et al. studies the solar radiation influence on temperature measurements by the Vaisala RS41 radiosonde using the Korean KRISS Upper Air Simulator environmental chamber.

The authors characterize the radiation error of this radiosonde as function of pressure, temperature, ventilation speed, and sensor orientation and tilt. They discuss the uncertainties of their measurements and show a rough comparison with the operational correction of this effect built into the Vaisala sounding system software.

The setup of their chamber and the measurements done as part of this study are excellent. However, the interpretation requires significant refinement, especially since they are providing a quantitative analysis of their measurements, which should eventually become applicable to sounding operations.

I would recommend publication of this manuscript only after major revisions, for which I provide detailed comments and suggestions below.

→ We thank the Reviewer for valuable comments. We have revised the manuscript to comply with the Reviewer's comment as below.

**Major comments:**

I will start from the end, since this is where this study potentially may have the most important significance.

**A) Application to real atmospheric observations and comparison to the Vaisala radiation correction table**

The largest weakness of this manuscript is that the measurements as such cannot be easily applied to sounding operations, since a substantial amount of interpretation of the measurements is missing. This is best demonstrated by the comparison with the Vaisala operational radiation correction table, which is presented, but discussed in only one sentence.

The Vaisala radiation correction table is applied in a significant fraction of soundings globally. The study here has the potential to support or improve this table. Unfortunately, this is not done and the reader is unable to decide, whether any of the measurements that were presented are in contradiction or support of the Vaisala table, or what factors need to be considered to make such a comparison. To be able to do so, some elements of the comparison have to be clarified. First, the solar angle used by Vaisala is not the same as the incident angle used by the authors. This needs to be clarified. Using their measurements, it should be possible to create a table similar to that by Vaisala. This requires describing the tilt of the sensor boom on the radiosonde in operational use

and making some assumptions about the pendulum and rotation movement of the sonde and applying their measurements to those. Second, the solar flux requires much more discussion. The radiation correction depends on the total flux, not just the direct solar flux. Some discussion about albedo and cloud cover is essential before a comparison can be done. It may not be possible to provide a direct validation of the Vaisala table, but at least the factors that contribute must be described in much more detail. The authors very briefly mention an effective solar flux and should expand this discussion greatly.

→ First of all, the users of RS41 cannot apply the radiation correction obtained by any method including this work, because the manufacturer does not provide raw temperature without correction. This work primarily intends to present the capability of the UAS to obtain a radiation correction of any radiosonde. RS41 is used as an example.

The tilting experiment in this work is to show that the radiation correction of RS41 is proportional to the effective irradiance ($S_{\text{eff}}$) to the sensor boom, not to the direct irradiance ($S_0$). The maximum tilting angle is 27 ° due to the small space in the test chamber. (We agree that larger tilting angles are needed to better imitate the actual effective irradiance to the sensor at different solar angles.) Nevertheless, the final radiation correction formula is provided based on the effective irradiance in Eq. (11).

We think that the calculation of the effective irradiance to the sensor and the application of the correction formula to actual soundings are the share of users because the situation is not identical globally. Nevertheless, an approach to the calculation of effective irradiance by averaging over the rotation is added in Fig. 8. Then, a graphical comparison of radiation corrections by the manufacturer and the UAS which includes the albedo effect is added in Fig. 9.

**Before:** The radiation correction of RS41 by the UAS is based on Eqs. (13) and (14) for different pressure ranges. Although the conditions for the UAS correction are different from those considered by the manufacturer, a rough comparison of the radiation corrections is presented in Table 11. For the UAS correction, the solar irradiance is assumed to be S = 1360 W·m$^{-2}$ at all pressure values. Depending on the effective irradiance ($S_{\text{eff}}$), the UAS correction value should be revised in a proportional manner using Eqs. (13) and (14).

**After (Line 397-413):** In order to apply the correction formula to actual soundings, the effective irradiance to the sensor should be known. However, radiosondes constantly change positions with respect to the solar irradiation through rotation and pendulum motion, the calculation of effective irradiance resorts on the mean of effective irradiance over the motion of radiosondes. Figure 8(a) shows a schematic diagram of a radiosonde with parameters that affect the effective irradiance $S_{\text{eff}}$ on the sensor. Then, the effective irradiance to the sensor can be calculated as follows:

$$S_{\text{eff}} = S_{\text{dir}} \cdot \left| \cos\alpha \cos\Theta \cos\varphi - \sin\Theta \sin\alpha) \right| , \tag{18}$$

$S_{\text{dir}}$ is solar direct irradiance, $\theta$ is boom tilting angle, $\alpha$ is solar elevation angle and $\varphi$ is azimuthal angle. The effective irradiation area ($A_{\text{eff}}/A_0$) on the sensor boom is averaged over rotation ($\varphi$) with a fixed tilting angle $\theta = 45$ ° and plotted as a function of the solar elevation angle as shown in Fig. 8(b). Using this effective irradiance, the radiation correction by the UAS is obtained and compared

with that of the manufacturer at two different $\alpha$ (45 ° and 90 °) as shown in Fig. 9. For the UAS correction, the solar direct irradiance is assumed to be 1360 W·m$^{-2}$ at all pressure values. To simulate the albedo effect, the radiation correction with additional irradiance of 400 W·m$^{-2}$ is also calculated. Consequently, the radiation correction of the UAS is smaller than the Vaisala by about 0.5−0.7 °C at −70 °C and 5 hPa when only the solar direct irradiance (1360 W·m$^{-2}$) is considered with the solar elevation angle $\alpha$ = 45−90 °. When the albedo effect is additionally included (400 W·m$^{-2}$), the gap between the two corrections is reduced to 0.04−0.4 °C at −70 °C and 5 hPa with $\alpha$ = 45−90 °. The radiation corrections of the manufacturer and the UAS at some representative conditions are summarized in **Table 11**.

**Modified Table (Table 11): Table 11** is modified to include the radiation correction of the UAS obtained by the above method.

**B) Uncertainties and their interpretation**

The uncertainty discussion is very important, but can be much improved. Table 4 is just an overview of the measurement ranges and a little confusing here. It could be deleted without loss. The discussion of the uncertainty in pressure and temperature measurement can be deleted as well. As Table 9 shows, this uncertainty does not contribute to the final result, which is immediately obvious given the weak dependence of the radiation error as function of pressure and temperature.

→ As mentioned in the above comment, this work aims at presenting the capability of the UAS to obtain a radiation correction of any radiosonde using RS41 as an example. The uncertainty of temperature and pressure should be presented in this perspective. The weak dependence of $T$ and $P$ in Table 9 also provides information to readers.

The uncertainty in the ventilation speed requires more discussion. The table lists a stability, but does not define to what it refers. It also lists a spatial gradient without specifying over which distance it applies. Most importantly, some discussion about the flow regime would be very useful. For laminar flows such as are more likely at low pressures, there should be significant velocity gradients from the walls inward. This is not discussed. If present, such gradients could explain the tilt results shown in Figure 6.

The uncertainty in irradiance lists the spatial gradient as the largest source. Again, gradient over what distance is meant here? It could be mentioned that the uncertainty of the radiation source is a negligible contribution compared to the lack of knowledge of the radiation field in true atmospheric soundings. This could be contrasted. In their conclusion, the authors indicate that they are working on a two-thermistor measurement with different emissivities. This discussion should be expanded and reference to the multi-thermistor work done by Schmidlin and others could be provided for reference.

→ Description on the measurement by the laser Doppler velocimetry (LDV) inside the test chamber is added. The central region around the sensor (35 mm x 35 mm) in the test chamber was measured by the LDV and thus the gradient from the wall was not measured.

The work by Schmidlin is mentioned with the dual-thermistor measurement in **Conclusions.**

**Added sentence (Line 103-104):** Thus, the reference value and SI traceability of the ventilation speed are obtained by using the sonic nozzles in the UAS.

**Before:** The generated air flow is measured through laser Doppler velocimetry to investigate the spatial gradient in the test chamber.

**After (Line 108-111):** The generated air flow is measured through laser Doppler velocimetry (LDV) (Dantec, Model: BSA F60) to investigate the spatial gradient in the test chamber. Ar-ion laser (3W) having a wavelength of 514.5 nm is used with a focal length of 400.1 mm and nominal beam spacing of 33 mm.

**Before:** The spatial gradient of the ventilation speed in the test chamber is measured through laser Doppler velocimetry at KRISS.

**After (Line 327-332):** The spatial gradient of the ventilation speed in the test chamber is measured through the LDV at KRISS. The measurement dimension using the LDV was 35 mm x 35 mm around the sensor (central) location with 5 mm interval (49 points) in the test chamber (50 mm x 50 mm). The measurement was performed at the condition of $v = 4.67$ m·s$^{-1}$ (reference value), $P = 550$ hPa, and room temperature. The LDV value averaged over the measurement area (35 mm x 35 mm) was 4.63 m·s$^{-1}$. The difference between the reference and the measurement average is assumed to have a rectangular probability distribution for the calculation of the uncertainty of spatial gradient.

**Added Statement (Line 341-342):** The uncertainty of the solar simulator will be negligible compared to that of the actual radiation field in atmospheric soundings due to the lack of knowledge.

**Before:** The temperature difference in the two sensors of the radiosonde is used to measure solar irradiance in situ.

**After (Line 435-439):** The temperature difference in the two sensors of the radiosonde is recorded with varying environmental parameters in the UAS to be reversely used to measure solar irradiance in situ during sounding. In this sense, the approach based on dual sensors is different from previous works that estimate the air temperature using several other temperatures measured by sensors with different emissivity (Schmidlin et al., 1986).

The uncertainty due to sensor rotation seems to be larger than the measured signal, which questions the uncertainty derivation; in particular since there clearly is a signal present.

→ The uncertainty is corrected to be the half of the previous value.

**Before:** the corresponding standard uncertainty ($k = 1$) is obtained considering the maximum value (0.06 °C) divided by $\sqrt{3}$. Consequently, the uncertainty due to sensor rotation is 0.035 °C ($k = 1$).

**After (Line 352-354):** the corresponding standard uncertainty ($k = 1$) is obtained considering the half-maximum value (0.03 °C) divided by $\sqrt{3}$. Consequently, the uncertainty due to sensor rotation is 0.017 °C ($k = 1$).

The final uncertainty (Section 4.10) is most likely pressure dependent, but no such pressure dependence is given. Whether or not there should be a pressure dependence would certainly require some more explanation.

→ The pressure and other conditions for the final uncertainty in **Table 10** are presented in **Table 9**. The uncertainty will be similar for other pressures and temperatures due to the weak dependency.

**C) Rotation and tilt discussion**

The discussion of rotation and tilt is rather sparse and could provide much more detail. The authors do not mentioned that the sensor boom of the Vaisala radiosonde is tilted from the vertical in operational use, which justifies their tilt measurements. I assume the 27 deg tilt used refers to the tilt of the sensor boom in operational use, but this has not been said. The tilt in real world soundings also depends on the pendulum motion of the radiosonde such as described by Dirksen et al. (2014). Thus, the sensor tilt is a little more complicated.

→ As mentioned in the above comment, the tilting experiment in this work is to show that the radiation correction of RS41 is proportional to the effective irradiance ($S_{\text{eff}}$) to the sensor boom, not to the direct irradiance ($S_0$). The maximum tilting angle is 27 ° due to the small space in the test chamber. An approach to the calculation of effective irradiance by averaging over the rotation is added in Fig. 8.

It is also important to point out, that the sensing element of the Vaisala radiosonde is a lengthy device, where tilt and rotation are likely to play an important role. Other radiosondes using spherical bead thermistors would be much less affected by tilt and rotation.

→ The difference depending on the sensor geometry is mentioned as suggested.

**Added Statement (Line 131-133):** The geometry of the temperature sensor of the Vaisala RS41 is a rod shape and thus the rotation and tilt affect the effective irradiance and the direction of air ventilation. Other radiosondes using spherical bead thermistors would be less affected by the rotation and tilt.

Measuring the temperature variation during rotation at 5 s is questionable, when the resolution of the data system is at best 1 s. Could it be that the minimal change at this speed is due to the inability to resolve the variations in time?

→ The ($T_{\text{on\_max}} - T_{\text{on\_min}}$) for 5 s duration is (0.01–0.02 °C) which is around the measurement resolution of RS41 (0.01 °C).

**Before**: The difference between the peaks ($T_{\text{on\_max}} - T_{\text{on\_min}}$) increases with the rotation period.

**After (Line 267-268)**: The difference between the peaks ($T_{on\_max} - T_{on\_min}$) for 5 s duration is (0.01–0.02 °C) which is around the measurement resolution of RS41 (0.01 °C) but is increased with the rotation period.

The authors speculate that mostly conduction from the sensor boom is responsible for the temperature variations during rotation. This is a reasonable assumption, but may require some more explanation and possibly an additional figure showing the geometry during rotation. Since the actual temperature sensor is in parallel with the axis of rotation, no change in surface area is expected here. However, the exposed surface area of the sensor boom changes significantly, which justifies the assumption. This should be shown explicitly. The temperature increase due to conduction appears to be small compared to the temperature increase due to direct irradiation of the sensor. This could be expanded as well.

→ The rotation axis is the temperature sensor itself, not the center of the boom. Therefore, the temperature sensor only spins on the spot and thus the distance between the sensor and the light source does not change at all during the rotation. This implies that the irradiance to the sensor is constant whereas the light incident angle (effective irradiance) to the sensor boom changes with rotation. This explains why the maximum temperature peak appears twice during a full cycle.

Based on the observation, the contribution of the conduction to $\Delta T_{rad}$ compared to that by the direct irradiation of the sensor is mentioned as suggested.

**Added Statement (Line 263-265):** The rotation axis is the temperature sensor itself, not the centre of the boom in this work. Therefore, the temperature sensor only spins on the spot and thus the distance between the sensor and the solar simulator does not change during the rotation.

**Added Statement (Line 280-282):** The relatively small ($T_{on\_max} - T_{on\_min}$) with respect to $\Delta T_{rad}$ observed in this work suggests that the contribution of the thermal conduction to $\Delta T_{rad}$ is small compared to that by the direct irradiation of the sensor.

**D) Underlying physical model**

All fit equations have the form shown in Figure 1. Is there a physical model that justifies this equation? If not, then this fit equation may be not be the most suitable, since it forces a split of the measurements into two pressure regimes. Using a single 5$^{th}$ order (or even 3$^{rd}$ order) polynomial of delta T over LOG P could provide a single fit over the entire pressure range from 5 to 500 hPa with similar results.

In addition, the fit equation provides a constant radiation error between 500 and 1000 hPa, which is somewhat surprising and in contrast to the model underlying the Vaisala table. A polynomial fit could improve here.

The polynomial fit would retain the temperature dependence at low pressures, which is an interesting result of their study.

→ Exponential fittings are replaced by polynomial fittings with $\log_{10} P$ to provide a single fit over entire pressure range of 5−500 hPa as suggested. Consequently, residuals by polynomial fittings (±0.03 °C) are smaller than those by exponential fittings (±0.04 °C) as shown in Fig. 2(b). Therefore, polynomial fittings are newly used to obtain radiation correction formula throughout the revised manuscript.

**Modified Figure (Figure 2):** Figure 2(a) is replotted by using parabola fittings of $\log_{10} P$ and the residual in Fig. 2(b) is newly obtained accordingly.

**Modified Equation (Equation 1, 9, 10 & 11):** The backbone of the Equations is changed to $\Delta T_{rad} = A_0(T) + B_0(T) \cdot \log(P) + C_0(T) \cdot [\log(P)]^2$

**Modified Table (Table 1)**: The original **Table 1** is removed and **Table 2** is relabeled to **Table 1**. **Table 1** is modified to include information on new coefficients of $A_0(T)$, $B_0(T)$, and $C_0(T)$.

**Minor comments:**

The authors could also make a statement how they expect the radiation correction to behave at speeds lower than 4 m/s. Some research groups fly radiosondes at lower ascent speeds to gain higher vertical resolution and would be interested in seeing the effect of the slower ascent.

→ The validity of the effect of the ventilation speed is limited to 4−7 m·s$^{-1}$ in this work. When $v$ is higher than 7 m·s$^{-1}$ or lower than 4 m·s$^{-1}$, the formula underestimates the correction value. This point is described in the revised manuscript.

**Added Statement (Line 249-250):** The linear relationship between the ventilation speed and the radiation correction in Eq. (9) is only valid in the range of 4−7 m·s$^{-1}$. When $v$ is higher than 7 m·s$^{-1}$ or lower than 4 m·s$^{-1}$, the formula underestimates the correction value.

In the introduction, the authors should also mention that the first approach to reducing the solar heating effect is applying highly reflective coatings. This is particularly relevant, since they later refer to thermistors with different emissivities.

→ The use of highly reflective coatings in previous works is mentioned in the Introduction.

**Before:** To minimize the effect of radiative heating of radiosonde temperature sensors, the size of sensors has been reduced.

**After (Line 47-48):** To minimize the effect of radiative heating of radiosonde temperature sensors, the size of sensors has been reduced (De Podesta et al., 2018) and highly reflective coatings are used (Luers and Eskridge, 1995; Schmidlin et al., 1986).

The authors could provide a discussion about the homogeneity of the temperature in their system, in particular the wall temperatures versus the air temperature, given that they have a very strong heat source.

→ The true radiation correction is the temperature difference between the sensor and air ($T_{on} -$ $T_{air}$). However, the air temperature measured in the current chamber system does not represent that in free atmosphere since the air is heated by irradiation for a short time while passing through the test section. The test section is also slightly heated by the irradiation and thus affects the sensor for the air temperature measurement in the chamber. In this work, the radiation correction ($\Delta T_{rad}$) is obtained by the difference in the temperatures with irradiation ($T_{on}$) and without irradiation ($T_{off}$); $\Delta T_{rad} = T_{on} - T_{off}$.

**Added Statement (Line 143-146):** The true radiation correction is the temperature difference between the sensor with irradiation and air ($T_{on} - T_{air}$). However, the air temperature measured in the current chamber system does not represent that in free atmosphere since the air is heated by irradiation for a short time while passing through the test section. The test section is also slightly heated by the irradiation and thus may affect the sensor installed for the air temperature measurement.

In Section 3.5, it is not perfectly clear, whether the authors varied the radiative flux or not. I assume that they did not, but rather do make the argument that the results should scale with the flux. This is reasonable, but could be made a little clearer.

→ The irradiance is fixed at $S_0 = 980$ W·m$^{-2}$ throughout this work. A sentence is added to argue that the result should be scaled with the flux.

**Added Sentence (Line 259):** The radiation correction ($\Delta T_{rad}$) is then scaled with the actual irradiance ($S$) by the factor of $S/S_0$.

The use of the factor SQRT(3) is mentioned repeatedly, but never justified. A reference to GUM would be useful with a brief explanation of what this factor does. This should be done only once at the beginning of the uncertainty section and other repetitions could be deleted.

→ The factor SQRT(3) is explained with the GUM as a reference

**Added Sentence (Line 332-333):**   Then, the standard uncertainty of this estimate is the half-width of the distribution divided by $\sqrt{3}$ (Iso, 2008).

**Added Reference:** GUM is added as a reference.

The arrows in Figures 1a and 1b should be a lighter color to make them better visible. The photo in Figure 1a could be brightened. The direction of the airflow should be indicated.

→ The color of the arrows in Figure 1 is changed. The direction of the air flow is indicated in Fig. 1(b).

In the legend of Figures 2 through 4, for simplification, the symbols and dashed lines should be combined (e.g.: --o--). The caption could then explain that the dashed lines show the fit and the symbols the actual measurements.

→ Combining the legend using (e.g.: --o--) is impossible because the symbols and fitting lines are generated separately using the Origin software.

Figure 2: If a $5^{th}$ order polynomial was used, then panels a) and b) could be combined to show the results over the entire pressure range using a logarithmic pressure axis.

→ Exponential fittings are replaced by polynomial fittings with $\log_{10} P$ to provide a single fit over entire pressure range of 5−500 hPa as suggested.

Figure 4: The indications of the different slopes should be removed from the actual Figure and the values could be added in a brief discussion either in the caption or in the main text.

→ The indications of the different slopes are removed from the Figure since they are already described in the manuscript.

Figure 6: The fat arrows should be removed. What they are supposed to indicate could be explained in the caption.

→ The fat arrows are removed from the Figure.

The language could be checked by a native speaker for more unusual expressions used by the authors.

→ The original manuscript was checked by a professional language editor before submission.

---

## Author Response (AR2)

**Referee #1**

**General Comment:** I want to thank the authors for addressing the points raised after the first review. This has greatly improved the quality of the manuscript.

→ We thank the Reviewer for the valuable comments to improve the quality of the manuscript.

**Major comments**

Section 3.2 shows in a convincing way the mechanism behind the influence of the ambient temperature on the radiation error. However, when making the comparison between the measurements and the theoretical calculation as in the plot in Fig 3b, the uncertainties in the data should be shown in the plot and taken into account when making statements about the agreement between measurements and calculations.

→ The experimental data in Fig. 3(b) is displayed by the mean and the standard deviation (of three repeated experiments). Since the expanded uncertainty of $\Delta T_{rad}$ is 0.1 °C at $k = 2$ (as shown in Table 9), the theoretical value is roughly consistent with the experimental value within the uncertainty.

**Before:** The radiation correction ($T_s - T_a$) at $T_a = 20$ °C and $-70$ °C is calculated by Eq. (5) and displayed together with the experimental values as shown in Fig. 3(b).

**After (Page 7, Line 192-193):** The radiation correction ($T_s - T_a$) at $T_a = 20$ °C and $-70$ °C is calculated by Eq. (5) and displayed together with the experimental values (mean and standard deviation of three repeated experiments) as shown in Fig. 3(b).

**Added statement (Page 7, Line 197-199):** The theoretical value is roughly consistent with the experimental value within the uncertainty of $\Delta T_{rad}$ (0.1 °C) as obtained in Section 4.9.

**l277-282:** The temperature oscillations due to the radiosonde rotation reported by von Rohden 2021 (AMT-187) can't be attributed to changes in distance to the lightsource. In the inset of Fig 2 of von Rohden 2021, it can be seen that the radiosonde is mounted such that the temperature sensor aligns with the axis of rotation, so that the distance variations are of the order of 1-2cm. This leads to ~3% variations of the flux on the temperature sensor, which is not sufficient to cause temperature variations of 0.3K. Therefore, the reason for the difference between the rotational temperature error dependence observed by von Rohden 2021 and reported in this paper still is not clear. Although it is mentioned that so-called raw data is used in the analysis: the data in the plot of Fig 6a appear to be 2 digits only.

→ In the work of von Rohden *et al.*, although the sensor is aligned with the rotation axis, the distance variation of the sensor boom is bigger than this work. If the distance variation of the sensor is not the reason, the effect of the sensor boom (distance variation) would be responsible for the observed difference between two works. This may explain the appearance of the maximum peak once (von Rohden *et al*) or twice (this work) in a full cycle. The previous manuscript was written in this context and we would like to keep this discussion.

**l405-415:** The plot in Fig 9 shows the comparison of the Vaisala correction and the correction based on the UAS experiments. In addition to the discussion of the quantitative differences at low pressures, the authors should comment on the qualitative difference between the shape of both curves.

→ We have added qualitative statements on the curves of radiation correction.

**Added statement (Page 17, Line 419-423):** Since solar direct irradiance (1360 W·m$^{-2}$) and additional diffuse irradiance (400 W·m$^{-2}$) are applied for all pressures, the radiation correction of this work can be exaggerated at high pressures. The radiation correction of the UAS is smaller than that of the manufacturer at low pressures, which is consistent with the recent finding using an independent laboratory setup. In the work of von Rohden *et al.*, the radiation correction was smaller than the manufacturer's by 0.35 K at 35 km (von Rohden et al., in review, 2021).

**Minor (mostly language-related) comments**

**l69:** 'the temperature effect" is unspecific. Suggested rephrasing: the influence of the ambient temperature on the radiation error was not investigated.

→ It is rephrased as suggested.

**Before:** the temperature effect on the sensors was not considered

**After (Page 3, Line 69-70):** the influence of the ambient temperature on the radiation error was not investigated.

**l143:** It is important to distinguish between the radiation error and the correction for this error. The radiation error is the difference between the measured temperature and the real air temperature (which is approximated by Ton - Tair). In the correction, an estimate of the radiation error is subtracted from the measured temperature.

→ Radiation error and the correction of the error are distinguished.

**Before:** The true radiation correction is the temperature difference between the sensor with irradiation and air ($T_{on} - T_{air}$).

**After (Page 5, Line 143):** Radiation error is the temperature difference between the sensor with irradiation and air ($T_{on} - T_{air}$).

**l145:** the heating of air by the absorption of visible radiation is negligible in this case. Do you perhaps mean the heating of the walls of the experiment chamber by the radiation? Do you have an estimate of the magnitude of this effect?

→ Although we have measured the temperature (a few centimeters) below the window using an independent thermometer, no significant response to the radiation state is observed. In general, the temperature is continuously increased by a few tens of mK while repeating the experiments three times for 10 min.

**Before:** The test section is also slightly heated by the irradiation and thus may affect the sensor installed for the air temperature measurement. The temperature rise due to irradiation ($\Delta T_{rad}$) is defined as the difference in the temperatures with irradiation ($T_{on}$) and without irradiation ($T_{off}$) as previously reported.

**After (Page 6, Line 145-149):** It is difficult to measure true air temperature at a shaded area in the test chamber using an independent thermometer because the test section is also slightly heated by the irradiation. The temperature measured below the window is continuously increased by a few tens of mK while repeating the experiments for 10 min. Thus, the radiation correction value ($\Delta T_{rad}$) is obtained by the difference in the temperatures with irradiation ($T_{on}$) and without irradiation ($T_{off}$) as previously reported.

**l153:** rapid -> enhanced. Regardless of temperature -> for all measured temperatures.

→ Changed as suggested.

**Before:** rapid

**After (Page 6, Line 155):** enhanced

**Before:** regardless of temperature

**After (Page 6, Line 155-156):** for all measured temperatures

**l158:** with -> in terms of

→ The word is replaced.

**Before:** with

**After (Page 6, Line 160):** in terms of

**l166:** insert comma after reduces.

→ A comma is inserted.

**Before:** reduces especially

**After (Page 6, Line 168)**: reduces, especially

**l180:** similarly with - > similar to

→ The phrase is changed.

**Before:** similarly with

**After (Page 7, Line 182 & 197)**: similar to

**Referee #2**

**Summary:**

The authors have revised their manuscript substantially and addressed most of my major concerns. However, a few points remain, on which the authors should elaborate.

→ We thank the Reviewer for the opportunity to further improve our manuscript. To comply with the Reviewer comments, the manuscript is revised as below.

**Comments:**

**A) Application to real atmospheric observations and comparison to the Vaisala radiation correction table**

The authors have included a discussion about steps required to transfer their laboratory measurements to atmospheric observations and have discussed their comparison with the radiation correction applied by Vaisala. The authors can show a qualitative agreement with the correction by Vaisala and are able to discuss the assumptions they had to make in this comparison.

→ As mentioned by the Reviewer, we have included the steps (assumptions) required to transfer our laboratory measurements to atmospheric observations in the previous revision. The mean over rotations of a radiosonde was used to calculate the effective irradiance to the sensor boom. The effect of surface albedo was also incorporated in addition to the solar direct irradiation. Using these approaches, a quantitative comparison with the manufacturer's correction table was made in **Figure 9** and **Table 11**.

**B) Uncertainties and their interpretation**

Although the authors have expanded their discussion about the flow speed in the test chamber, they have not addressed my question about the flow regime or the possibility of gradients in the test chamber. They explain that they measured 49 points with a spacing of 5 mm, but only provide an average of these measurements. It should be easy to make a statement whether significant gradients were observed and what the flow regime most likely is. My guess is that the flow regime should be turbulent, but the authors should make this statement and justify it.

→ As the Reviewer mentioned, the flow regime is turbulent because Reynolds number is high ($\sim 10^5$) at the experimental condition of $P = 550$ hPa and room temperature. In the test chamber, 49 points on the same plane were measured one by one using the LDV and the standard deviation was 0.47 ms$^{-1}$. Although the flow rate of the outermost points tends to be smaller than others, no significant spatial gradient is observed. This may be because the outermost point is spaced 10 mm apart from the walls of the test chamber.

**Before:** The measurement dimension using the LDV was 35 mm x 35 mm around the sensor (central) location with 5 mm interval (49 points) in the test chamber (50 mm x 50 mm).

**After (Page 13, Line 331-333):** The measurement dimension using the LDV was 30 mm x 30 mm around the sensor (central) location with 5 mm interval (49 points). Thus, the outermost measurement points were spaced 10 mm apart from the walls of the test chamber (50 mm x 50 mm).

**Added statement (Page 13, Line 334):** The flow regime is turbulent because Reynolds number is high ($\sim 10^5$) at this experimental condition.

**Before:** The average by the LDV over the entire measurement area was 4.63 m·s$^{-1}$.

**After (Page 13, Line 334-335):** The average and the standard deviation by the LDV over the entire measurement area were 4.63 m·s$^{-1}$ and 0.47 m·s$^{-1}$, respectively.

**Added statement (Page 13, Line 335-337):** Although the flow rate of the outermost points tends to be smaller than others, no significant spatial gradient is observed. This may be because the spacing (10 mm) between the outermost measurement points and the walls of the test chamber.

The argument for the uncertainty estimate in section 4.6 is in contradiction with the same argument in section 4.7 and 4.8. The reason for the sqrt(3) is the assumption that the uncertainty is equally distributed in the a particular range. That would argue for using the maximum value, not the half-maximum and scaling it by sqrt(3) to make it Gaussian equivalent. This argument should be applied consistently throughout. Section 3.6: The uncertainty of the rotation is still close to the actually measured signal, which appears unlikely.

→ The reason of the half-maximum used in section 4.6 is that ($T_{\text{on\_max}} - T_{\text{on\_min}}$) is about double of ($T_{\text{on\_max}} - T_{\text{on}}$) or ($T_{\text{on}} - T_{\text{on\_min}}$) which is equivalent to the maximum in section 4.7 and 4.8.

**Added statement (Page 15, Line 359-360):** The reason of using the half-maximum is that ($T_{\text{on\_max}} - T_{\text{on\_min}}$) is about double of ($T_{\text{on\_max}} - T_{\text{on}}$) or ($T_{\text{on}} - T_{\text{on\_min}}$).

Lines 273ff: Whether the effect of rotation is temperature dependent is speculation and not supported by the measurements provided. However, the effect of rotation is small, so this discussion becomes largely irrelevant.

→ As the Reviewer mentioned, the temperature dependence of rotation effect will be largely irrelevant because the effect of rotation is small.

Lines 277ff: The main reason, why the measurements of Rohden differ is that in their setup the axis of rotation is not parallel to the sensor. Therefore, the exposed cross section changes in their setup. The authors may highlight their statement that thermal conduction from the sensor boom is likely very small. I believe this is the essential result of this section.

→ The statement on the thermal conduction is highlighted.

**Before:** The relatively small ($T_{\mathrm{on\_max}} - T_{\mathrm{on\_min}}$) with respect to $\Delta T_{\mathrm{rad}}$ observed in this work suggests that the contribution of the thermal conduction to $\Delta T_{\mathrm{rad}}$ is small compared to that by the direct irradiation of the sensor.

**After (Page 11, Line 283-285):** It should be highlighted that the relatively small ($T_{\mathrm{on\_max}} - T_{\mathrm{on\_min}}$) with respect to $\Delta T_{\mathrm{rad}}$ observed in this work suggests that the contribution of the thermal conduction to $\Delta T_{\mathrm{rad}}$ is small compared to that by the direct irradiation of the sensor.

**C) Rotation and tilt discussion**

The authors have substantially expanded the discussion about rotation and tilt of the sonde in flight. This makes is simpler to translate their results to atmospheric observations.

→ We thank the Reviewer for his/her valuable comments to expand our discussion on the rotation and tilting of the radiosonde.

**D) Underlying physical model**

Using the polynomial fit simplifies the presentation of the data and their discussion. It avoids a speculation about an underlying physical model.

→ We thank the Reviewer for suggesting a polynomial fitting in the previous revision. After using the fitting, the presentation is clearer than the former version.